# Landmark-Guided Policy Optimization for Multi-Objective Language Model Selection

## Abstract

Selecting a pretrained large language model (LLM) to fine-tune for a task-specific dataset can be time-consuming and costly. With several candidate models available to choose from, varying in size, architecture, and pretraining data, finding the best often involves extensive trial and error. In addition, the "best" model may not necessarily be the one with the lowest test loss, as practical considerations such as deployment costs, inference throughput, and limited search budgets might also play crucial roles. To address this, we introduce LAMPS (LAnguage Model Pareto Selection), a novel and open-source multi-objective AutoML framework that quickly identifies near-Pareto-optimal pretrained LLMs for a task-specific dataset. It is based on two key ideas: (1) landmark fine-tuning, which generates early performance indicators of the candidate models, and (2) meta-learning via reinforcement learning, which learns an effective selection policy from historical performance data (a meta-dataset). Our results show that, on held-out datasets, LAMPS reduces search time by an average of 73%71% compared to exhaustive search, while still covering more than 99%98% of the optimal target space hypervolume.

## 1 Introduction

Fine-tuning a pretrained large language model (LLM) on task-specific datasets is currently the dominant paradigm for achieving state-of-the-art performance in several natural language processing (NLP) tasks (Radford et al., 2019), including question answering (Chowdhery et al., 2023), machine translation (Raffel et al., 2020), summarization (Aghajanyan et al., 2020), and classification (Yang, 2019). However, different pretrained models yield varying downstream performance due to differences in size, architecture, pretraining data, and other intrinsic factors. Therefore, as the set of available pretrained models is already extensive, the important question arises: How can we efficiently find the best model for a task-specific dataset?

A common practice in NLP is to select the largest available model, driven by the belief that larger models invariably provide better performance (e.g., accuracy, F1, perplexity, cross-entropy, depending on the downstream task). Although this is generally true, several studies have shown that smaller models are comparable to or even outperform larger ones for specialized tasks (Ouyang et al., 2022; Sanh et al., 2020; Hoffmann et al., 2022; Wahba et al., 2023; DeepSeek-AI et al., 2025; Wang et al., 2025). Moreover, in real-world scenarios, always choosing larger models inevitably leads to higher operational costs and greater environmental impact. This underscores the need to incorporate additional factors into the model selection process beyond a single task-specific performance metric.

A multi-objective perspective is, then, essential to capture the broader spectrum of trade-offs that practitioners face when selecting pretrained LLMs for fine-tuning. In the absence of better alternatives, practitioners may turn to exhaustive search. Although theoretically sound, this method quickly becomes prohibitively expensive for a large number of candidate models, especially for target datasets with several million examples. As language models continue to expand in scale and diversity, there is an increasing need for a principled, holistic, and efficient selection strategy, especially with the growing interest in specialized LLM-based AI agents (Gutowska, 2024; Ma et al., 2024).

In this paper, we introduce **LAMPS** (**LA**nguage Model **P**areto **S**election), a novel and open-source multi-objective AutoML framework for selecting LLMs to fine-tune on task-specific datasets. It

integrates two complementary strategies: (1) *landmark fine-tuning*, which generates early performance indicators for candidate models by evaluating them on incrementally larger subsets of the training data; and (2) *meta-learning via reinforcement learning*, which leverages historical model performance data on multiple datasets to learn how to efficiently allocate training resources for new datasets. In other words, this process generates a policy that manages the selection and early stopping of candidate models, adjusting its strategy based on both observed and historical performance to efficiently discard low-potential models and prioritize promising ones.

Our main contributions are as follows: (i) Formulating the language model selection for fine-tuning explicitly as a multi-objective optimization problem; (ii) Introducing LAMPS, a novel and open-source AutoML framework combining landmark fine-tuning, meta-learning, and reinforcement learning to rapidly identify near-Pareto-optimal language models for a new task-specific dataset.

The remainder of the paper is organized as follows. Section 2 reviews relevant related work. Section 3 states the multi-objective optimization problem. The method is proposed in Section 5 and Section 6 presents the experimental setup and main findings. We conclude in Section 7.

## 2 RELATED WORK

Selecting an appropriate base learner (model, algorithm, pipeline, etc.) for a given task has been a long-standing research topic and is usually called *model selection* (Bozdogan, 1987; Maron & Moore, 1993; McQuarrie & Tsai, 1998; Chapelle et al., 2002; Biem, 2003; Brazdil et al., 2003; Zhao & Yu, 2006; Adankon & Cheriet, 2009). Among the different approaches available, meta-learning has been a popular choice (Kalousis & Hilario, 2000; Fürnkranz et al., 2002; Brazdil & Giraud-Carrier, 2018; Jain et al., 2024; de Amorim et al., 2025; Farhadi et al., 2025), mainly due to its ability to transfer knowledge from prior learning experiences, reducing the cost of exploration and improving sample efficiency.

In this section, we provide a brief overview of the related areas that form the foundation of our LAMPS framework.

**Pretrained Model Selection in Deep Learning**  Fine-tuning pretrained deep learning models for specific downstream tasks has become the standard approach in both computer vision and natural language processing. Compared to training from scratch, fine-tuning is far more efficient and requires much less data than pretraining (Hepburn, 2018). For this reason, being able to select the right pretrained model efficiently is becoming increasingly relevant due to the considerable computational costs and the rapid introduction of new models with varying sizes, architectures, training data, and capabilities. To the best of our knowledge, the only work that explicitly addresses the selection of LLMs for fine-tuning is by Monteiro et al. (2024), but it neither considers the multi-objective aspects of the model selection nor adjusts its recommendations based on actual fine-tuning learning curves.

**Subsampling Landmarks**  A sampling landmark is a performance-based meta-feature, representing the performance of a particular model on samples of available data, providing a quick estimate of its performance (Brazdil et al., 2022; Pfahringer et al., 2000) and, consequently, allowing indirect characterization of the target dataset. One variant is called *subsampling landmarks*, which considers a sequence of sample sizes in increasing order, effectively representing the early stages of the learning curve (Soares et al., 2001; Fürnkranz & Petrak, 2001). This is conceptually related to the scaling laws observed in deep neural networks (Kaplan et al., 2020) and large language models (Zhang et al., 2024), which describe the predictable relationship between model performance and, among other factors, dataset size. Subsampling landmarks can thus be viewed as a localized and practical proxy for these scaling behaviors, enabling performance forecasting without requiring full-scale training. Similar ideas have been applied for hyperparameter optimization (Domhan et al., 2015; Jamieson & Talwalkar, 2016; Klein et al., 2017; Li et al., 2018), which use partial learning curves to stop training poor configurations early. Such methods, however, remain inherently single-objective and cannot directly address the multi-objective settings considered in this work.

**Multi-Task and Meta-Reinforcement Learning**  Reinforcement learning is a powerful tool for sequential decision-making problems, but it often struggles with generalization to new (unknown)

tasks, requiring large amounts of data to readapt effectively. Two areas address these limitations: multi-task reinforcement learning (MTRL) (Teh et al., 2017; Sodhani et al., 2021) and meta-reinforcement learning (Meta-RL) (Finn et al., 2017; Nichol et al., 2018; Wang et al., 2024). MTRL trains a single policy across a distribution of tasks, leveraging shared structure to improve generalization and learning efficiency. In contrast, Meta-RL focuses on learning a policy that can rapidly adapt to new tasks using limited data, typically by encoding task-specific information into its internal state or parameters. In this work, we focus on MTRL, as our goal is to evaluate policies on previously unseen datasets without further adaptation at test time.

**Multi-Objective Reinforcement Learning**  Multi-objective reinforcement learning (MORL) extends standard RL by optimizing policies with respect to multiple, often conflicting objectives rather than a single reward. Prior research on MORL, often combined with meta-learning, has largely relied on scalarization or objective preferences, requiring weight sweeps across many preferences to approximate the Pareto front (Lu et al., 2024; Wang et al., 2024; Liu & Qian, 2021; Chen et al., 2019). Because each weight vector defines a different scalar objective, changing preferences generally requires another sweep (i.e., additional fine-tuning runs), so computation grows with each revision. By contrast, we target Pareto coverage in a single, efficient run.

**Hyperparameter Optimization**  Work in hyperparameter optimization (HPO), often overlapping with neural architecture search (NAS), frequently leverages early training signals to discard low-promising configurations and reduce computational cost (Falkner et al., 2018; Li et al., 2020; Awad et al., 2021; Wistuba et al., 2022). Standard HPO methods, however, are fundamentally single-objective, and extending them to multi-objective settings typically relies on scalarization. As shown by Schmucker et al. (2021), scalarization-based adaptations usually perform significantly worse than methods explicitly designed for multi-objective search, highlighting a key limitation of conventional HPO techniques in scenarios requiring Pareto-efficient model selection.

## 3 PROBLEM STATEMENT

Consider a target dataset $\mathcal{D}$ and a set $\mathcal{X}$ of candidate pretrained language models to be fine-tuned. Then, given $n$ metrics of interest (objectives), the problem can be formulated as the following multi-objective optimization problem:

$$\min_{x \in \mathcal{X}} \quad \big(f_1(x, \mathcal{D}), \dots, f_n(x, \mathcal{D})\big)$$
$$\text{s.t.} \quad f_i(x, \mathcal{D}) \leq f_i^{\max} \quad \text{for all } i = 1, \dots, n, \tag{1}$$

where $f_i(x, \mathcal{D})$ represents the value of the $i$-th objective function after fine-tuning the pretrained model $x \in \mathcal{X}$ on the task-specific dataset $\mathcal{D}$, and $f_i^{\max}$ denotes an arbitrary upper bound for that objective.

Common objectives may include final test loss, training time (cost), inference throughput, number of model parameters, and resource usage (i.e., number of GPUs). It is very common that some objectives conflict with each other. For example, achieving a lower test loss may require longer training time or more GPUs. For this reason, there is typically no single solution that is optimal across all objectives. Hence, the notion of optimality is based on Pareto-dominance, or simply dominance, as defined below.

**Definition 1** (Weak dominance). *A solution $x_1 \in \mathcal{X}$ weakly dominates another solution $x_2 \in \mathcal{X}$, denoted $x_1 \succeq x_2$, if $f_i(x_1, \mathcal{D}) \leq f_i(x_2, \mathcal{D})$ for all $i \in \{1, \dots, n\}$. That is, $x_1$ is not worse than $x_2$ in all objectives.*

**Definition 2** (Pareto-dominance). *A solution $x_1 \in \mathcal{X}$ dominates another solution $x_2 \in \mathcal{X}$, denoted $x_1 \succ x_2$, if $f_i(x_1, \mathcal{D}) \leq f_i(x_2, \mathcal{D})$ for all $i \in \{1, \dots, n\}$, with at least one of these inequalities holding strictly. That is, there is $j \in \{1, \dots, n\}$ such that $f_j(x_1, \mathcal{D}) < f_j(x_2, \mathcal{D})$. In other words, $x_1$ dominates $x_2$ if $x_1$ is not worse than $x_2$ in all objectives, but it is better in at least one of them.*

**Definition 3** (Pareto-optimal). *A model $x^* \in \mathcal{X}$ is Pareto-optimal if there is no other $x \in \mathcal{X}$ that dominates $x^*$.*

One way to evaluate and compare sets of candidate solutions is to use the *hypervolume indicator* (Guerreiro et al., 2021; Emmerich et al., 2005), which quantifies the volume of the objective space weakly dominated by a set of solutions and bounded above by a given reference point

$r = [f_1^{\max}, \ldots, f_n^{\max}]^\top$. For any subset $X \subset \mathcal{X}$, the hypervolume indicator is denoted as $H_{\mathcal{D}}(X, r)$. Intuitively, each solution in $X$ defines a box in the objective space, with one corner at the objective values of the solution and the opposite corner at the reference point $r$. It is defined formally as follows:

**Definition 4** (Hypervolume indicator). *Given a set of points $S \subset \mathbb{R}^n$ and a reference point $r \in \mathbb{R}^n$, the hypervolume indicator of $S$ is the measure of the region weakly dominated by $S$ and bounded above by $r$, i.e.,*

$$H(S, r) = \Lambda \left( \bigcup_{\substack{p \in S \\ p \leq r}} [p, r] \right),$$

*where $\Lambda(\cdot)$ denotes the Lebesgue measure and $[p, r] = \{q \in \mathbb{R}^n \mid \forall i = 1, \ldots, n : p_i \leq q_i \leq r_i\}$ denotes the box delimited below by $p \in S$ and above by $r$.*

It has been shown that maximizing the hypervolume indicator is equivalent to finding the Pareto optimal set (Guerreiro et al., 2021; Liu et al., 2019). Figure 1 illustrates this with a practical comparison, showing that the Pareto-optimal set has the highest hypervolume. Thus, the problem in (1) can be reformulated as a single-objective problem as follows:

$$\max_{X \subset \mathcal{X}} \quad H_{\mathcal{D}}(X, r) \tag{2}$$

A trivial solution would involve fine-tuning all models on the target dataset (i.e., $X = \mathcal{X}$), but this is computationally intractable. To encourage computational efficiency, we introduce a regularization term penalizing the number of selected pretrained models:

$$\max_{X \subset \mathcal{X}} \quad H_{\mathcal{D}}(X, r) - \lambda |X| \tag{3}$$

where $\lambda > 0$ is a user-defined penalty factor. To ensure that the optimal solution for the problem in Equation 3 contains exactly all Pareto-optimal solutions, $\lambda$ must satisfy the following theorem, proved in Appendix H:

**Theorem 1** (Condition on $\lambda$). *The optimal solution $X^* \subset \mathcal{X}$ of problem 3 contains only and exactly all Pareto-optimal solutions if and only if:*

$$0 < \lambda \leq \min_{x \in X^*, X \subseteq X^*} \Delta H_{\mathcal{D}}(x \mid X), \tag{4}$$

*where $\Delta H_{\mathcal{D}}(x \mid X)$ denotes the incremental hypervolume obtained by adding the Pareto-optimal solution $x$ to the subset $X \subseteq X^*$.*

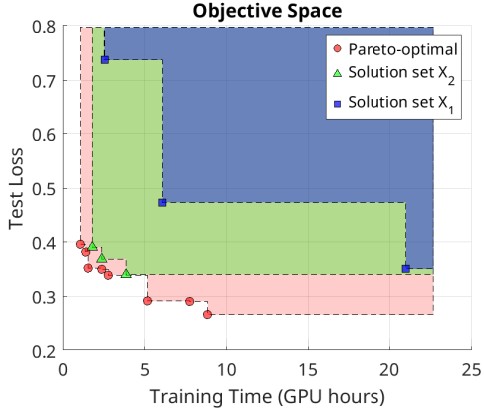

Figure 1: Illustration of the hypervolume indicator in a bi-objective setting, corresponding to the shaded areas. Set $X_2$ yields a larger hypervolume than $X_1$, which is closer to the true Pareto front.

In other words, the penalty $\lambda$ must be smaller than or equal to the smallest incremental hypervolume gained by including a new Pareto-optimal solution into the subset of selected solution candidates. If this condition holds, the optimal solution set will include only all Pareto-optimal solutions. The next sections present empirical strategies for quickly providing near-Pareto optimal solutions.

## 4    LANDMARK FINE-TUNING

Fine-tuning a pretrained model on a task-specific dataset is inevitable if one desires to evaluate its true performance and determine its suitability for a given application. However, as discussed earlier, evaluating every candidate model is computationally expensive. Prior work on hyperparameter optimization suggests that evaluating models for only a single epoch can already be a good proxy for its final performance (Egele et al., 2023). However, training for just one epoch may still consume significant resources, particularly for large models and datasets.

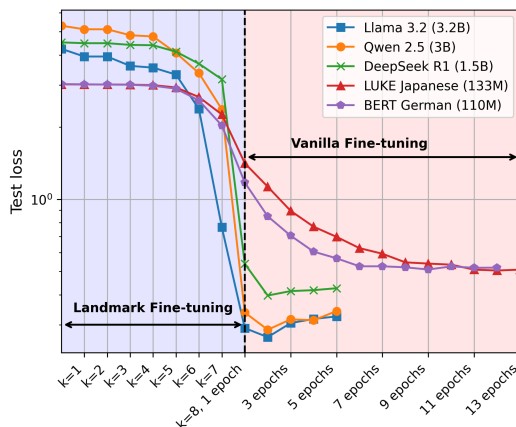

Figure 2: Landmark fine-tuning on the 20 Newsgroups dataset using $K = 8$. Larger models start off worse but eventually outperform smaller ones. Notably, models that improve quickly early on tend to achieve lower final loss, suggesting that the initial segments of the learning curves can help predict the overall performance.

To mitigate this inefficiency and allow for even earlier identification of unpromising candidates, we propose *landmark fine-tuning*, a lightweight fine-tuning strategy based on subsampling landmarks to obtain early estimates of objective values $f_i(x, \mathcal{D})$ for $i \in \{1 \dots n\}$.

Given that the target dataset $\mathcal{D}$ has a training and a test split, namely $\mathcal{D}^{\text{train}}$ and $\mathcal{D}^{\text{test}}$, the core idea is to split $\mathcal{D}^{\text{train}}$ into $K$ exponentially larger subsets $\mathcal{D}_1 \dots \mathcal{D}_K$. Each subset $\mathcal{D}_k$ contains $\left\lfloor \frac{1}{2^{(K-k)}} |\mathcal{D}^{\text{train}}| \right\rfloor$ samples, where $\mathcal{D}_k \subset \mathcal{D}_{k+1}$ for $k = 1 \dots K - 1$.

The process starts by fine-tuning a pretrained model on $\mathcal{D}_1$ for a *single epoch* and evaluating it on the entire $\mathcal{D}^{\text{test}}$. Next, it continues the fine-tuning process on the subsequent (larger) subset $\mathcal{D}_2$, repeating this process up to $\mathcal{D}_K$ (100% of the training dataset). After that, we continue fine-tuning the model for more epochs until convergence or other stop criterion.

Figure 2 shows a practical example of landmark fine-tuning with $K = 8$, depicting the learning curves (test cross-entropy loss) of five different pretrained models fine-tuned on the 20 Newsgroups dataset. Two non-English LLMs are included to illustrate the performance of less suitable models on an English dataset. Notice that larger models start with higher losses than smaller ones, but eventually overtake them, achieving lower final losses. In addition, among the larger models, those that improve more quickly in the initial steps tend to achieve better final test loss. These observations support the idea that early segments of the training curve can indeed be predictive of final loss, with predictions becoming more accurate as additional curve segments are provided.

## 5 META-LEARNED RESOURCE ALLOCATION VIA REINFORCEMENT LEARNING

Although landmark fine-tuning provides early performance estimates, it is still necessary to determine when to continue training a candidate model or not, based on partial information collected so far. To address this, we train a reinforcement learning agent on a meta-dataset of historical fine-tuning trajectories, covering a diverse set of pretrained LLMs and downstream tasks. The agent learns to allocate training resources by tracking how performance evolves across landmark steps, enabling fast and generalizable identification of near-Pareto-optimal models.

**Observation space**  The observation space defines the information available to the RL agent at each decision step. At each time step $t$, the RL agent observes, for every candidate model, the objectives of interest (e.g., the elapsed training time and test loss), together with the number of fine-tuning steps that each candidate has completed.

**Action space** The action space specifies the set of decisions available to the RL agent at each step. For each time step $t$, the agent selects an action $a_t \in \{1, \ldots, m\}$, representing the index of a candidate pretrained model, where $m$ is the total number of candidates. Each action corresponds to allocating one additional fine-tuning step to the selected model. To improve exploration efficiency, we apply invalid action masking for terminated models (Huang & Ontañón, 2022). A binary mask specifies which models remain available for selection. The policy then samples only from this valid subset by setting the probability of invalid actions to zero. This prevents wasted trials on completed models and makes the exploration phase more efficient, as the agent can focus its decisions on candidates that may still yield improvements.

**Termination condition** An episode corresponds to the full search process and terminates when all Pareto-optimal models have been fully fine-tuned[1], thereby achieving the maximum hypervolume. This termination condition is only necessary during policy training, where the agent has access to privileged information that indicates when the Pareto frontier has been fully explored. Thanks to invalid action masking, the episode is guaranteed to terminate within a finite number of steps, preventing the agent from getting stuck in infinite allocations to unproductive models.

**Training algorithm** For training the policy, we adopted a standard multi-task reinforcement learning setup, in which a single policy is optimized jointly across all training tasks.~~Distral (distill and transfer learning), a framework for multi-task RL where the knowledge gained in one task is distilled into a shared policy, then transferred to other tasks via regularization using a Kullback-Leibler (KL) divergence.~~ As the underlying optimizer, we adopted Proximal Policy Optimization (PPO) (Schulman et al., 2017), which provides stable on-policy updates and performs reliably in multi-task settings.~~combining its stability with cross-task transfer from Distral.~~

**Rewards** The reward function links our multi-objective search problem to the policy's learning process. Let $X_t \subseteq \mathcal{X}$ be the set of fully fine-tuned models by time step $t$, and let $T$ be the length of the episode. Inspired by equation 3, we could initially define a sparse reward function

$$r_t = \begin{cases} H_{\mathcal{D}}(X_t) - \lambda|X_t| & \text{if } t = T \\ 0 & \text{otherwise} \end{cases}, \tag{5}$$

so that PPO would maximize

$$\max_\theta \quad \mathbb{E}_{\rho \sim \pi_\theta}\left[\sum_{t=0}^{T} \gamma^t r_t\right] = \mathbb{E}_{\rho \sim \pi_\theta}\left[H_{\mathcal{D}}(X_T) - \lambda|X_T|\right], \tag{6}$$

where $\rho$ is a trajectory sampled using policy $\pi_\theta$, and $\gamma$ is the discount factor.

Because an episode terminates only after all Pareto-optimal models have been fully fine-tuned, $H_{\mathcal{D}}(X_T)$ is identical for every trajectory and, therefore, constant. The objective thus collapses to minimizing the expected number of models evaluated, i.e., $\mathbb{E}[-|X_T|]$. Notice that $\lambda$ also vanishes in this sparse reward setting, so we do not need to estimate it. Finally, to make the reward positive and incentivize faster convergence to the optimal, we adopted the following sparse reward function:

$$r_t = \begin{cases} \frac{|\mathcal{X} \setminus X_t|}{\Delta_t} & \text{if } t = T \\ 0 & \text{otherwise} \end{cases}, \tag{7}$$

where $\Delta_t$ is the cumulative wall-clock time spent up to time step $t$. In other words, we seek to maximize the number of pretrained models not fully fine-tuned, divided by the time spent to find all Pareto-optimal models. This produces a positive and well-scaled learning signal and preserves the optimal solution of equation 3, as the highest reward is obtained when $X_T = X^*$. Appendix D presents additional evidence showing that the proposed reward signal in equation 7 leads PPO to converge to the optimal solution during the training phase.

---

[1] A model is considered fully fine-tuned when its validation loss stops improving for a fixed number of consecutive epochs.

---

**Algorithm 1:** LAMPS Search Procedure

---

**Input** : Training dataset $\mathcal{D}^{\text{train}}$, validation dataset $\mathcal{D}^{\text{val}}$
**Input** : Policy $\pi_\theta$
**Output:** Set of non-dominated fine-tuned models $\hat{X}^\star$

1 Initialize time step $t \leftarrow 0$;
2 Initialize the set of selected models $X \leftarrow \varnothing$;
3 Evaluate candidate models on $\mathcal{D}^{\text{val}}$ and construct the initial state $s_0$;
4 **while** *search budget not exhausted* **do**
5      Select action $a_t \leftarrow \arg\max_a \ \pi_\theta(a \mid s_t)$ ;
6      Fine-tune model $x_{a_t}$ for one additional fine-tuning step on $\mathcal{D}^{\text{train}}_{k+1}$;
7      Evaluate updated performance on $\mathcal{D}^{\text{val}}$;
8      **if** *stopping criterion met for model $x_{a_t}$* **then**
9          $\lfloor \ X \leftarrow X \cup \{x_{a_t}\}$
10     Update environment state $s_{t+1}$;
11     $t \leftarrow t + 1$;
12 $\hat{X}^\star = \{ x \in X \mid \nexists y \in X : y \succ x \}$;
13 **return** $\hat{X}^\star$;

---

**Meta-dataset** To meta-train a policy capable of efficiently identifying (or approximating) the Pareto-optimal set for new task-specific datasets, we conducted a fine-tuning campaign and constructed a meta-dataset containing fully recorded learning curves of 70 pretrained LLMs, each landmark fine-tuned on multiple datasets (see Appendix E). This setup enables the agent to query arbitrary trajectories during its training, allowing the use of on-policy algorithms such as PPO.

**Deployment (search procedure)** Given a trained policy $\pi_\theta$ and a target dataset $\mathcal{D}$, the search procedure of LAMPS is outlined in Algorithm 1. The process begins by constructing the initial state $s_0$ through zero-shot evaluation of all candidate models on the test split. It also serves as a sanity check to ensure that each model is available, downloaded properly, and compatible with the available hardware (and drivers) where the search will be performed. The policy then proceeds by selecting and executing new actions until the search budget is exhausted. In the end, dominated solutions are filtered out, so that only the best trade-offs are presented to the user.

## 6 EXPERIMENTS AND RESULTS

This section presents our experimental setup and main findings, demonstrating how well the trained policy generalizes to held-out datasets. The experiments presented in this section~~All experiments in this paper~~ were conducted on eight NVIDIA A100 (40 GB) GPUs.

### 6.1 EXPERIMENTAL SETUP

**Pretrained LLMs** We tested 70 different pretrained language models, spanning models from a few million parameters (ALBERT) to eight billion parameters (DeepSeek-R1). These models cover languages such as English, Japanese, Chinese, German, Dutch, Spanish, and many of which are multilingual. The complete list of pretrained models can be found in Appendix F. We did not considered any Mixture-of-Experts (MoE) models, as they are usually more challenging to fine-tune and more prone to overfitting (Fedus et al., 2022; Shen et al., 2024).

**Fine-tuning Setup** We adopted full-model fine-tuning, which updates all parameters of the pretrained models. Although parameter-efficient methods such as LoRA (Hu et al., 2022) or layer-freezing strategies can significantly reduce computational overhead, full fine-tuning often leads to better downstream performance (Zhang et al., 2024; Shuttleworth et al., 2024). All models were fine-tuned under identical hyperparameter settings. See Appendix C for details.

**Reinforcement Learning Setup**  We used the following libraries: Stable-Baselines3 (SB3) for PPO implementation and invalid action masking (Raffin et al., 2021), and the Gymnasium library for standardized environment definition (Towers et al., 2024).

**Objectives**  For the optimization criteria, we focus primarily on two objectives: validation loss (measured via cross-entropy) and model size (in number of parameters)~~training time required for completing the fine-tuning~~. Validation loss is a widely accepted proxy for task-specific performance, and model size~~training time~~ serves as a practical and measurable approximation for other metrics, such as VRAM~~model size~~, inference throughput, deployment cost, etc. These choices are not fixed for LAMPS, as the framework is objective-agnostic. Hence, any measurable objectives can be used[2], as long as the corresponding metrics are recorded in the meta-dataset. In Appendix B we show additional results for machine translation, considering two and three objectives.

**Reference point**  We set the reference point by taking the worst values of the chosen objectives across the meta-dataset and adding a 10% margin. This reference point is used only during policy training for computing the hypervolume. It is not required at test time when evaluating the trained policy (Algorithm 1).

**Baselines**  ~~To our knowledge, no prior work has explored the same multi-objective optimization problem. Hence, a direct comparison with other existing methods was not possible. For this reason,~~ We compared LAMPS with four~~three~~ ~~basic~~ baselines:

- **Blind**: chooses actions at random. Its performance serves as a lower bound on performance and represents the worst-case scenario.
- **Oracle**: assumes prior knowledge of the Pareto-optimal models for a given task. The performance of this approach represents the best-case scenario. In practice, this information is not available and serves only as a theoretical upper bound.
- **ZigZag**: a simple heuristic that sorts all candidate models by their number of parameters, then selects them in an alternating order (from largest to smallest and vice versa) in an attempt to quickly increase the covered hypervolume.
- **MO-ASHA**: multi-objective asynchronous successive halving, combined with $\epsilon$-net exploration strategy (Schmucker et al., 2021).

**Evaluation Method**  To evaluate LAMPS's generalization, we employed leave-one-out cross-validation (Hastie et al., 2009), where one dataset is held exclusively for testing. For each fold, the policy is trained on the remaining datasets for a fixed number of steps and then evaluated on the held-out dataset. This allows us to assess how well the learned policy transfers to previously unseen tasks. To ensure robustness, this procedure was repeated five times, and we report the average performance across these runs.

## 6.2 RESULTS

To evaluate the generalization of LAMPS to unseen datasets, Figure 3 reports the time required to reach 99%~~98%~~ of the optimal hypervolume in each held-out dataset. Recall that, in our problem formulation, achieving optimal hypervolume corresponds to identifying all Pareto-optimal models. For reference, we also include the time needed for an exhaustive search to complete. Across the twelve held-out tasks, LAMPS achieves the best performance in nine datasets (75%), whereas MO-ASHA wins only three (25%). Although MO-ASHA is the strongest baseline overall, its behavior is markedly less stable: on several datasets, its search time approaches the BLIND baseline, which never occurs with LAMPS.

To illustrate the practical implications, consider the Amazon dataset: running an exhaustive search on a single A100 40GB NVIDIA GPU ($3.67 hourly) would cost $5,141.67, whereas LAMPS

---

[2]Choosing only highly correlated objectives collapses the Pareto frontier, effectively reducing a multi-objective problem to a single-objective one. Since adding objectives increases search complexity, it is important to select conflicting and informative objectives to make the multi-objective formulation meaningful.

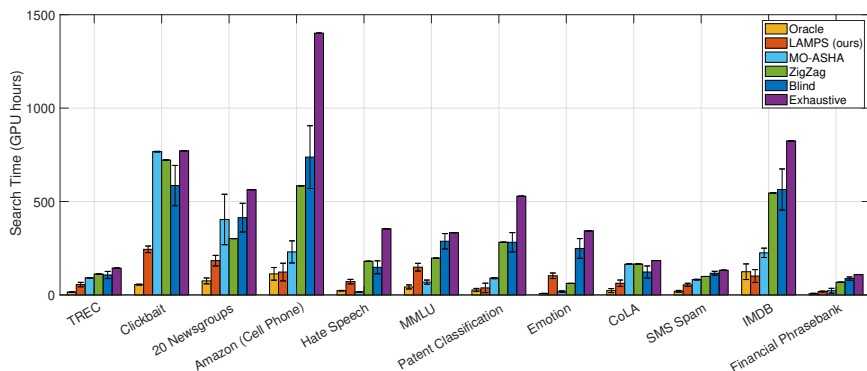

Figure 3: Mean time cost (in GPU hours) to reach 99%98% of the optimal hypervolume indicator on held-out datasets. For reference, we also show the time to complete an exhaustive search. On average, LAMPS reduces the search time by 73.6%71% compared to the exhaustive search, outperforming other feasible methods in 9 out of 12 datasets.and being comparable to the ORACLE in 7 out of 9 datasets.

reduces the cost to just $449.58 with only a 1% degradation in the hypervolume. The strongest competing baseline, MO-ASHA, would cost $845.20 to reach the same performance.

Figure 4 provides further insight by tracking the progression of the average hypervolume over search time. For comparability, hypervolume values are normalized by the maximum hypervolume, and we report the *hypervolume loss* (1−normalized hypervolume) in logarithmic scale to highlight when the policy reaches optimality. Although LAMPS does not always reach optimality in a timely manner (compared to the other baselines), it clearly achieves near-optimal solutions quickly, eventually faster than ORACLE. This ability to deliver high-quality solutions at a fraction of the cost makes LAMPS the best trade-off between efficiency and solution quality, positioning it as a pragmatic and strong tool for practitioners.

Moreover, in multi-objective applications, the end user must ultimately select a preferred solution from the Pareto front, often revisiting trade-offs as requirements, constraints, or business priorities. By quickly providing a diverse set of strong candidates, LAMPS not only accelerates the search, but also enables practitioners to reconsider or change their choice later without having to undergo another expensive search, offering both flexibility and long-term practical value.

## 7 CONCLUSION

We presented LAMPS, a novel and open-source AutoML framework for efficiently selecting pre-trained language models for fine-tuning, framing it as a multi-objective optimization problem. By combining landmark fine-tuning and meta-learning via reinforcement learning, LAMPS significantly reduces search costs while maintaining near-optimal performance. Experiments show that LAMPS reduces search time by 73%71% on average with minimal hypervolume degradation. To our knowledge, this is the first framework to deliver Pareto-efficient selection and fine-tuning for LLMs, establishing a new baseline for cost-aware AutoML and paving the way toward sustainable, high-performance deployment of foundation models.

## REFERENCES

Mathias M. Adankon and Mohamed Cheriet. Model selection for the ls-svm. application to handwriting recognition. *Pattern Recognition*, 42(12):3264–3270, 2009. ISSN 0031-3203. doi: https://doi.org/10.1016/j.patcog.2008.10.023. URL https://www.sciencedirect.com/science/article/pii/S0031320308004494. New Frontiers in Handwriting Recognition.

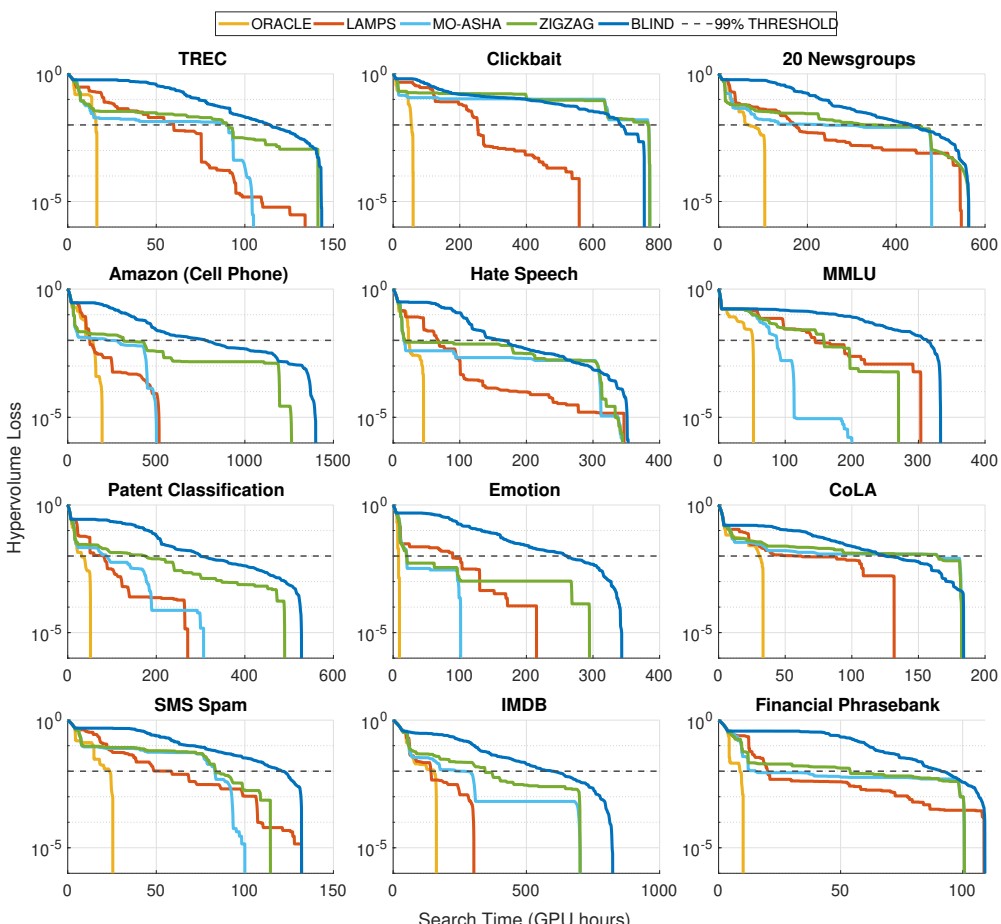

Figure 4: Evolution of the mean hypervolume indicator on held-out datasets as a function of search budget. LAMPS rapidly identifies near-optimal solutions (dashed line) in ~~nine~~seven out of twelve ~~nine~~ cases, demonstrating strong generalization capabilities, even when trained on a small meta-dataset.

Armen Aghajanyan, Akshat Shrivastava, Anchit Gupta, Naman Goyal, Luke Zettlemoyer, and Sonal Gupta. Better fine-tuning by reducing representational collapse. *arXiv preprint arXiv:2008.03156*, 2020.

Noor Awad, Neeratyoy Mallik, and Frank Hutter. Dehb: Evolutionary hyberband for scalable, robust and efficient hyperparameter optimization. In Zhi-Hua Zhou (ed.), *Proceedings of the Thirtieth International Joint Conference on Artificial Intelligence, IJCAI-21*, pp. 2147–2153. International Joint Conferences on Artificial Intelligence Organization, 8 2021. doi: 10.24963/ijcai.2021/296. URL https://doi.org/10.24963/ijcai.2021/296. Main Track.

Alain Biem. A model selection criterion for classification: application to hmm topology optimization. In *Proceedings of the Seventh International Conference on Document Analysis and Recognition - Volume 1*, ICDAR '03, pp. 104, USA, 2003. IEEE Computer Society. ISBN 0769519601.

Hamparsum Bozdogan. Model selection and akaike's information criterion (aic): The general theory and its analytical extensions. *Psychometrika*, 52(3):345–370, 1987. doi: 10.1007/BF02294361. URL https://doi.org/10.1007/BF02294361.

Pavel Brazdil and Christophe Giraud-Carrier. Metalearning and algorithm selection: progress, state of the art and introduction to the 2018 special issue. *Machine learning*, 107:1–14, 2018.

Pavel Brazdil, Jan N. van Rijn, Carlos Soares, and Joaquin Vanschoren. *Dataset Characteristics (Metafeatures)*, pp. 53–75. Springer International Publishing, Cham, 2022. ISBN 978-3-030-67024-5. doi: 10.1007/978-3-030-67024-5_4. URL https://doi.org/10.1007/978-3-030-67024-5_4.

Pavel B. Brazdil, Carlos Soares, and Joaquim Pinto Da Costa. Ranking learning algorithms: using ibl and meta-learning on accuracy and time results. *Mach. Learn.*, 50(3):251–277, March 2003. ISSN 0885-6125. doi: 10.1023/A:1021713901879. URL https://doi.org/10.1023/A:1021713901879.

Abhijnan Chakraborty, Bhargavi Paranjape, Sourya Kakarla, and Niloy Ganguly. Stop clickbait: Detecting and preventing clickbaits in online news media. In *Advances in Social Networks Analysis and Mining (ASONAM), 2016 IEEE/ACM International Conference on*, pp. 9–16. IEEE, 2016.

Olivier Chapelle, Vladimir Vapnik, and Yoshua Bengio. Model selection for small sample regression. *Machine Learning*, 48(1):9–23, 2002. doi: 10.1023/A:1013943418833. URL https://doi.org/10.1023/A:1013943418833.

Xi Chen, Ali Ghadirzadeh, Mårten Björkman, and Patric Jensfelt. Meta-learning for multi-objective reinforcement learning. In *2019 IEEE/RSJ International Conference on Intelligent Robots and Systems (IROS)*, pp. 977–983. IEEE Press, 2019. doi: 10.1109/IROS40897.2019.8968092. URL https://doi.org/10.1109/IROS40897.2019.8968092.

Aakanksha Chowdhery, Sharan Narang, Jacob Devlin, Maarten Bosma, Gaurav Mishra, Adam Roberts, Paul Barham, Hyung Won Chung, Charles Sutton, Sebastian Gehrmann, et al. Palm: Scaling language modeling with pathways. *Journal of Machine Learning Research*, 24(240): 1–113, 2023.

Thomas Davidson, Dana Warmsley, Michael Macy, and Ingmar Weber. Automated hate speech detection and the problem of offensive language. In *Proceedings of the 11th International AAAI Conference on Web and Social Media*, ICWSM '17, pp. 512–515, 2017.

Lucas B. V. de Amorim, George D. C. Cavalcanti, and Rafael M. O. Cruz. Meta-scaler: A meta-learning framework for the selection of scaling techniques. *IEEE Transactions on Neural Networks and Learning Systems*, 36(3):4805–4819, 2025. doi: 10.1109/TNNLS.2024.3366615.

DeepSeek-AI et al. Deepseek-r1: Incentivizing reasoning capability in llms via reinforcement learning, 2025. URL https://arxiv.org/abs/2501.12948.

Gianna M. Del Corso, Antonio Gullí, and Francesco Romani. Ranking a stream of news. In *Proceedings of the 14th International Conference on World Wide Web*, WWW '05, pp. 97–106, New York, NY, USA, 2005. Association for Computing Machinery. ISBN 1595930469. doi: 10.1145/1060745.1060764. URL https://doi.org/10.1145/1060745.1060764.

Tobias Domhan, Jost Tobias Springenberg, and Frank Hutter. Speeding up automatic hyperparameter optimization of deep neural networks by extrapolation of learning curves. In *Proceedings of the 24th International Conference on Artificial Intelligence*, IJCAI'15, pp. 3460–3468. AAAI Press, 2015. ISBN 9781577357384.

Romain Egele, Isabelle Guyon, Yixuan Sun, and Prasanna Balaprakash. Is one epoch all you need for multi-fidelity hyperparameter optimization? *arXiv preprint arXiv:2307.15422*, 2023.

Michael Emmerich, Nicola Beume, and Boris Naujoks. An emo algorithm using the hypervolume measure as selection criterion. In *Proceedings of the Third International Conference on Evolutionary Multi-Criterion Optimization*, EMO'05, pp. 62–76, Berlin, Heidelberg, 2005. Springer-Verlag. ISBN 3540249834. doi: 10.1007/978-3-540-31880-4_5. URL https://doi.org/10.1007/978-3-540-31880-4_5.

Stefan Falkner, Aaron Klein, and Frank Hutter. Bohb: Robust and efficient hyperparameter optimization at scale. In *International conference on machine learning*, pp. 1437–1446. PMLR, 2018.

Armin Farhadi, Roya Hatami, Mohammad Robat Mili, Christos Masouros, and Mehdi Bennis. A meta-learning approach for energy-efficient resource allocation and antenna selection in star-bd-ris aided wireless networks. *IEEE Wireless Communications Letters*, pp. 1–1, 2025. doi: 10.1109/LWC.2025.3543780.

William Fedus, Barret Zoph, and Noam Shazeer. Switch transformers: scaling to trillion parameter models with simple and efficient sparsity. *J. Mach. Learn. Res.*, 23(1), January 2022. ISSN 1532-4435.

Chelsea Finn, Pieter Abbeel, and Sergey Levine. Model-agnostic meta-learning for fast adaptation of deep networks. In *Proceedings of the 34th International Conference on Machine Learning - Volume 70*, ICML'17, pp. 1126–1135. JMLR.org, 2017.

Johannes Fürnkranz and Johann Petrak. An evaluation of landmarking variants. In *Working Notes of the ECML/PKDD 2000 Workshop on Integrating Aspects of Data Mining, Decision Support and Meta-Learning*, pp. 57–68. Citeseer, 2001.

Johannes Fürnkranz, Josef Petrak, Pavel Brazdil, and Carlos Soares. On the use of fast subsampling estimates for algorithm recommendation. Technical report, Austrian Research Institute for Artificial Intelligence, 2002.

Alec Go, Richa Bhayani, and Lei Huang. Twitter sentiment classification using distant supervision. *CS224N project report, Stanford*, 1(12):2009, 2009.

Andreia P. Guerreiro, Carlos M. Fonseca, and Luís Paquete. The hypervolume indicator: Computational problems and algorithms. *ACM Comput. Surv.*, 54(6), July 2021. ISSN 0360-0300. doi: 10.1145/3453474. URL https://doi.org/10.1145/3453474.

Anna Gutowska. What are AI agents?, July 2024. URL https://www.ibm.com/think/topics/ai-agents. Accessed: 2025-02-07.

Trevor Hastie, Robert Tibshirani, and Jerome Friedman. *The elements of statistical learning: data mining, inference, and prediction*. Springer, New York, 2 edition, 2009. ISBN 978-0-387-84857-0. URL https://hastie.su.domains/ElemStatLearn/.

Jason Hepburn. Universal language model fine-tuning for patent classification. In Sunghwan Mac Kim and Xiuzhen (Jenny) Zhang (eds.), *Proceedings of the Australasian Language Technology Association Workshop 2018*, pp. 93–96, Dunedin, New Zealand, December 2018. URL https://aclanthology.org/U18-1013/.

Jordan Hoffmann, Sebastian Borgeaud, Arthur Mensch, Elena Buchatskaya, Trevor Cai, Eliza Rutherford, Diego de las Casas, Lisa Anne Hendricks, Johannes Welbl, Aidan Clark, Tom Hennigan, Eric Noland, Katherine Millican, George van den Driessche, Bogdan Damoc, Aurelia Guy, Simon Osindero, Karen Simonyan, Erich Elsen, Oriol Vinyals, Jack William Rae, and Laurent Sifre. An empirical analysis of compute-optimal large language model training. In Alice H. Oh, Alekh Agarwal, Danielle Belgrave, and Kyunghyun Cho (eds.), *Advances in Neural Information Processing Systems*, 2022. URL https://openreview.net/forum?id=iBBcRUlOAPR.

Edward J Hu, yelong shen, Phillip Wallis, Zeyuan Allen-Zhu, Yuanzhi Li, Shean Wang, Lu Wang, and Weizhu Chen. LoRA: Low-rank adaptation of large language models. In *International Conference on Learning Representations*, 2022. URL https://openreview.net/forum?id=nZeVKeeFYf9.

Shengyi Huang and Santiago Ontañón. A closer look at invalid action masking in policy gradient algorithms. *The International FLAIRS Conference Proceedings*, 35, May 2022. ISSN 2334-0762. doi: 10.32473/flairs.v35i.130584. URL http://dx.doi.org/10.32473/flairs.v35i.130584.

Nishant Jain, Arun S. Suggala, and Pradeep Shenoy. Improving generalization via meta-learning on hard samples. In *2024 IEEE/CVF Conference on Computer Vision and Pattern Recognition (CVPR)*, pp. 27590–27599, 2024. doi: 10.1109/CVPR52733.2024.02606.

Kevin Jamieson and Ameet Talwalkar. Non-stochastic best arm identification and hyperparameter optimization. In Arthur Gretton and Christian C. Robert (eds.), *Proceedings of the 19th International Conference on Artificial Intelligence and Statistics*, volume 51 of *Proceedings of Machine Learning Research*, pp. 240–248, Cadiz, Spain, 09–11 May 2016. PMLR. URL https://proceedings.mlr.press/v51/jamieson16.html.

A. Kalousis and M. Hilario. Model selection via meta-learning: a comparative study. In *Proceedings 12th IEEE Internationals Conference on Tools with Artificial Intelligence. ICTAI 2000*, pp. 406–413, 2000. doi: 10.1109/TAI.2000.889901.

Jared Kaplan, Sam McCandlish, Tom Henighan, Tom B. Brown, Benjamin Chess, Rewon Child, Scott Gray, Alec Radford, Jeffrey Wu, and Dario Amodei. Scaling laws for neural language models. *OpenAI blog*, 2020. URL https://arxiv.org/abs/2001.08361.

Aaron Klein, Stefan Falkner, Simon Bartels, Philipp Hennig, and Frank Hutter. Fast Bayesian Optimization of Machine Learning Hyperparameters on Large Datasets. In Aarti Singh and Jerry Zhu (eds.), *Proceedings of the 20th International Conference on Artificial Intelligence and Statistics*, volume 54 of *Proceedings of Machine Learning Research*, pp. 528–536. PMLR, 20–22 Apr 2017. URL https://proceedings.mlr.press/v54/klein17a.html.

Ken Lang. 20 newsgroups dataset, 1995. URL http://qwone.com/~jason/20Newsgroups/. Accessed: 2025-02-26.

Liam Li, Kevin Jamieson, Afshin Rostamizadeh, Ekaterina Gonina, Jonathan Ben-tzur, Moritz Hardt, Benjamin Recht, and Ameet Talwalkar. A system for massively parallel hyperparameter tuning. In I. Dhillon, D. Papailiopoulos, and V. Sze (eds.), *Proceedings of Machine Learning and Systems*, volume 2, pp. 230–246, 2020. URL https://proceedings.mlsys.org/paper_files/paper/2020/file/a06f20b349c6cf09a6b171c71b88bbfc-Paper.pdf.

Lisha Li, Kevin Jamieson, Giulia DeSalvo, Afshin Rostamizadeh, and Ameet Talwalkar. Hyperband: A novel bandit-based approach to hyperparameter optimization. *Journal of Machine Learning Research*, 18(185):1–52, 2018. URL http://jmlr.org/papers/v18/16-558.html.

Fei-Yu Liu and Chao Qian. Prediction guided meta-learning for multi-objective reinforcement learning. In *2021 IEEE Congress on Evolutionary Computation (CEC)*, pp. 2171–2178. IEEE Press, 2021. doi: 10.1109/CEC45853.2021.9504972. URL https://doi.org/10.1109/CEC45853.2021.9504972.

Yang Liu, Jingxuan Wei, Xin Li, and Minghan Li. Generational distance indicator-based evolutionary algorithm with an improved niching method for many-objective optimization problems. *IEEE Access*, 7:63881–63891, 2019. doi: 10.1109/ACCESS.2019.2916634.

Junlin Lu, Patrick Mannion, and Karl Mason. A meta-learning approach for multi-objective reinforcement learning in sustainable home environments. In *European Conference on Artificial Intelligence (ECAI) 2024*, January 2024.

Wei Ma, Daoyuan Wu, Yuqiang Sun, Tianwen Wang, Shangqing Liu, Jian Zhang, Yue Xue, and Yang Liu. Combining fine-tuning and llm-based agents for intuitive smart contract auditing with justifications. *arXiv preprint arXiv:2403.16073*, 2024.

Andrew L. Maas, Raymond E. Daly, Peter T. Pham, Dan Huang, Andrew Y. Ng, and Christopher Potts. Learning word vectors for sentiment analysis. In Dekang Lin, Yuji Matsumoto, and Rada Mihalcea (eds.), *Proceedings of the 49th Annual Meeting of the Association for Computational Linguistics: Human Language Technologies*, pp. 142–150, Portland, Oregon, USA, June 2011. Association for Computational Linguistics. URL https://aclanthology.org/P11-1015/.

P. Malo, A. Sinha, P. Korhonen, J. Wallenius, and P. Takala. Good debt or bad debt: Detecting semantic orientations in economic texts. *Journal of the Association for Information Science and Technology*, 65, 2014.

Oded Maron and Andrew W. Moore. Hoeffding races: accelerating model selection search for classification and function approximation. In *Proceedings of the 7th International Conference on Neural Information Processing Systems*, NIPS'93, pp. 59–66, San Francisco, CA, USA, 1993. Morgan Kaufmann Publishers Inc.

Allan D R McQuarrie and Chih-Ling Tsai. *Regression and time series model selection*. WORLD SCIENTIFIC, 1998. doi: 10.1142/3573. URL https://www.worldscientific.com/doi/abs/10.1142/3573.

Marcio Monteiro, Charu Karakkaparambil James, Marius Kloft, and Sophie Fellenz. Characterizing text datasets with psycholinguistic features. In Yaser Al-Onaizan, Mohit Bansal, and Yun-Nung Chen (eds.), *Findings of the Association for Computational Linguistics: EMNLP 2024*, pp. 14977–14990, Miami, Florida, USA, November 2024. Association for Computational Linguistics.

Alex Nichol, Joshua Achiam, and John Schulman. On first-order meta-learning algorithms, 2018. URL https://openai.com/index/on-first-order-meta-learning-algorithms/.

Long Ouyang, Jeffrey Wu, Xu Jiang, Diogo Almeida, Carroll Wainwright, Pamela Mishkin, Chong Zhang, Sandhini Agarwal, Katarina Slama, Alex Ray, John Schulman, Jacob Hilton, Fraser Kelton, Luke Miller, Maddie Simens, Amanda Askell, Peter Welinder, Paul F Christiano, Jan Leike, and Ryan Lowe. Training language models to follow instructions with human feedback. In S. Koyejo, S. Mohamed, A. Agarwal, D. Belgrave, K. Cho, and A. Oh (eds.), *Advances in Neural Information Processing Systems*, volume 35, pp. 27730–27744. Curran Associates, Inc., 2022. URL https://proceedings.neurips.cc/paper_files/paper/2022/file/b1efde53be364a73914f58805a001731-Paper-Conference.pdf.

Bernhard Pfahringer, Hilan Bensusan, and Christophe G. Giraud-Carrier. Meta-learning by landmarking various learning algorithms. In *Proceedings of the Seventeenth International Conference on Machine Learning*, ICML '00, pp. 743–750, San Francisco, CA, USA, 2000. Morgan Kaufmann Publishers Inc. ISBN 1558607072.

Alec Radford, Jeffrey Wu, Rewon Child, David Luan, Dario Amodei, Ilya Sutskever, et al. Language models are unsupervised multitask learners. *OpenAI blog*, 1(8):9, 2019.

Colin Raffel, Noam Shazeer, Adam Roberts, Katherine Lee, Sharan Narang, Michael Matena, Yanqi Zhou, Wei Li, and Peter J. Liu. Exploring the limits of transfer learning with a unified text-to-text transformer. *J. Mach. Learn. Res.*, 21(1), January 2020. ISSN 1532-4435.

Antonin Raffin, Ashley Hill, Adam Gleave, Anssi Kanervisto, Maximilian Ernestus, and Noah Dormann. Stable-baselines3: Reliable reinforcement learning implementations. *Journal of Machine Learning Research*, 22(268):1–8, 2021. URL http://jmlr.org/papers/v22/20-1364.html.

Victor Sanh, Lysandre Debut, Julien Chaumond, and Thomas Wolf. Distilbert, a distilled version of bert: smaller, faster, cheaper and lighter, 2020. URL https://arxiv.org/abs/1910.01108.

Robin Schmucker, Michele Donini, Muhammad Bilal Zafar, David Salinas, and Cédric Archambeau. Multi-objective asynchronous successive halving, 2021. URL https://arxiv.org/abs/2106.12639.

John Schulman, Filip Wolski, Prafulla Dhariwal, Alec Radford, and Oleg Klimov. Proximal policy optimization algorithms, 2017. URL https://arxiv.org/abs/1707.06347.

Sheng Shen, Le Hou, Yanqi Zhou, Nan Du, Shayne Longpre, Jason Wei, Hyung Won Chung, Barret Zoph, William Fedus, Xinyun Chen, Tu Vu, Yuexin Wu, Wuyang Chen, Albert Webson, Yunxuan Li, Vincent Y Zhao, Hongkun Yu, Kurt Keutzer, Trevor Darrell, and Denny Zhou. Mixture-of-experts meets instruction tuning: A winning combination for large language models. In *The Twelfth International Conference on Learning Representations*, 2024. URL https://openreview.net/forum?id=6mLjDwYte5.

Reece Shuttleworth, Jacob Andreas, Antonio Torralba, and Pratyusha Sharma. Lora vs full fine-tuning: An illusion of equivalence. *arXiv preprint arXiv:2410.21228*, 2024.

Carlos Soares, Johann Petrak, and Pavel Brazdil. Sampling-based relative landmarks: systematically test-driving algorithms before choosing. In Pavel Brazdil and Alípio Jorge (eds.), *Progress in Artificial Intelligence*, pp. 88–95, Berlin, Heidelberg, 2001. Springer Berlin Heidelberg. ISBN 978-3-540-45329-1.

Shagun Sodhani, Amy Zhang, and Joelle Pineau. Multi-task reinforcement learning with context-based representations. In Marina Meila and Tong Zhang (eds.), *Proceedings of the 38th International Conference on Machine Learning*, volume 139 of *Proceedings of Machine Learning Research*, pp. 9767–9779. PMLR, 18–24 Jul 2021. URL https://proceedings.mlr.press/v139/sodhani21a.html.

Yee Whye Teh, Victor Bapst, Wojciech Marian Czarnecki, John Quan, James Kirkpatrick, Raia Hadsell, Nicolas Heess, and Razvan Pascanu. Distral: robust multitask reinforcement learning. In *Proceedings of the 31st International Conference on Neural Information Processing Systems*, NIPS'17, pp. 4499–4509, Red Hook, NY, USA, 2017. Curran Associates Inc. ISBN 9781510860964.

Jörg Tiedemann. Parallel data, tools and interfaces in OPUS. In Nicoletta Calzolari, Khalid Choukri, Thierry Declerck, Mehmet Uğur Doğan, Bente Maegaard, Joseph Mariani, Asuncion Moreno, Jan Odijk, and Stelios Piperidis (eds.), *Proceedings of the Eighth International Conference on Language Resources and Evaluation (LREC'12)*, pp. 2214–2218, Istanbul, Turkey, May 2012. European Language Resources Association (ELRA). URL http://www.lrec-conf.org/proceedings/lrec2012/pdf/463_Paper.pdf.

Mark Towers, Ariel Kwiatkowski, Jordan Terry, John U Balis, Gianluca De Cola, Tristan Deleu, Manuel Goulão, Andreas Kallinteris, Markus Krimmel, Arjun KG, et al. Gymnasium: A standard interface for reinforcement learning environments. *arXiv preprint arXiv:2407.17032*, 2024.

Yasmen Wahba, Nazim Madhavji, and John Steinbacher. A comparison of svm against pre-trained language models (plms) for text classification tasks. In Giuseppe Nicosia, Varun Ojha, Emanuele La Malfa, Gabriele La Malfa, Panos Pardalos, Giuseppe Di Fatta, Giovanni Giuffrida, and Renato Umeton (eds.), *Machine Learning, Optimization, and Data Science*, pp. 304–313, Cham, 2023. Springer Nature Switzerland. ISBN 978-3-031-25891-6.

Qi Wang, Chengwei Zhang, and Bin Hu. Dynamic programming with meta-reinforcement learning: a novel approach for multi-objective optimization. *Complex & Intelligent Systems*, 10(4): 5743–5758, 2024. doi: 10.1007/s40747-024-01469-1. URL https://doi.org/10.1007/s40747-024-01469-1.

Shangshang Wang, Julian Asilis, Ömer Faruk Akgül, Enes Burak Bilgin, Ollie Liu, and Willie Neiswanger. Tina: Tiny reasoning models via lora, 2025. URL https://arxiv.org/abs/2504.15777.

Martin Wistuba, Arlind Kadra, and Josif Grabocka. Supervising the multi-fidelity race of hyperparameter configurations. *Advances in Neural Information Processing Systems*, 35:13470–13484, 2022.

Zhilin Yang. Xlnet: Generalized autoregressive pretraining for language understanding. *arXiv preprint arXiv:1906.08237*, 2019.

Biao Zhang, Zhongtao Liu, Colin Cherry, and Orhan Firat. When scaling meets LLM finetuning: The effect of data, model and finetuning method. In *The Twelfth International Conference on Learning Representations*, 2024. URL https://openreview.net/forum?id=5HCnKDeTws.

Peng Zhao and Bin Yu. On model selection consistency of lasso. *Journal of Machine Learning Research*, 7(90):2541–2563, 2006. URL http://jmlr.org/papers/v7/zhao06a.html.

## A  LAMPS: GETTING STARTED

This section demonstrates how to use LAMPS for a new, unseen dataset. The provided policy was meta-trained across the datasets described in the main paper for a total of 10M~~45M~~ steps, minimizing the following objectives: validation loss and model size~~training time~~. Before running it, make sure to have sufficient disk space (at least 2TB) for intermediate storage of models and checkpoints. In addition, some models hosted on Hugging Face may require license agreements or explicit acceptance terms. Ensure that the necessary access is granted to your user account prior to execution.

Listing 1: Running LAMPS for a new dataset.

```
# Create the Python environment
conda create -n lamps python=3.10
conda activate lamps

# Install dependencies
pip install -r requirements.txt

# Initiate the search using the trained policy
python eval.py --policy "policies/ALL-MTRL-30M_steps.zip" \
   --dataset "stanfordnlp/imdb" \
   --input-col "text" \
   --target-col "label"
```

## B  ADDITIONAL EXPERIMENTS: MACHINE TRANSLATION

To further assess the generality, objective-agnosticism, and scalability of LAMPS, we conducted additional experiments in the domain of machine translation. Using 4x NVIDIA A100 (40GB) GPUs, we constructed a meta-dataset comprising 38 translation directions from the OPUS Books corpus, a collection of copyright-free literary texts spanning a wide range of languages (Tiedemann, 2012).

### B.1  TWO OBJECTIVES

Table 1 reports the time (in GPU hours) to recover 99.9% of the optimal hypervolume when optimizing for model size and evaluation loss. The results show that LAMPS transfers meta-learned knowledge effectively to the majority of held-out task-specific datasets, being comparable to the ORACLE in 32 out of 38 cases.

### B.2  THREE OBJECTIVES

To evaluate how well LAMPS scales to higher-dimensional objective spaces, we extend our analysis to a three-objective setting involving model size, evaluation loss, and BLEU score. As shown in Table 2, the ORACLE requires substantially more time to recover 99.9% of the optimal hypervolume than in the 2D case. This increase reflects the expansion of the Pareto frontier (now a surface) when BLEU is added, making the search space more challenging to find or approximate.

Despite this increased complexity, LAMPS remains the strongest overall method by a large margin, achieving the best search performance on 27 of 38 datasets (71%). Although MO-ASHA becomes more competitive in this 3D setting, increasing its win rate to 10/38 (26%), its performance also becomes significantly less stable: on several language pairs, it drops to the level of the BLIND baseline, which has not been observed in the any of the 2D experiments. This widening performance gap indicates that LAMPS scales more reliably and consistently as the dimensionality of the objective space increases.

Table 1: Time in GPU-hours to recover 99.9% of the optimal hypervolume on held-out datasets, optimizing for two objectives: model size and evaluation loss.

| Dataset | Oracle | LAMPS (ours) | MO-ASHA | ZigZag | Blind | Exhaustive |
|---|---|---|---|---|---|---|
| DE-EN | $46.0_{\pm6.0}$ | $91.1_{\pm5.9}$ | $\mathbf{50.7_{\pm5.9}}$ | 88.2 | $121.2_{\pm3.2}$ | 123.9 |
| DE-ES | $18.2_{\pm0.0}$ | $\mathbf{18.2_{\pm0.0}}$ | $35.5_{\pm5.6}$ | 53.1 | $71.3_{\pm2.4}$ | 73.6 |
| DE-FR | $18.4_{\pm0.1}$ | $\mathbf{18.3_{\pm0.1}}$ | $36.4_{\pm2.3}$ | 66.1 | $90.4_{\pm3.0}$ | 93.0 |
| DE-IT | $20.7_{\pm0.0}$ | $\mathbf{20.7_{\pm0.0}}$ | $38.0_{\pm1.8}$ | 57.7 | $76.0_{\pm1.7}$ | 77.1 |
| DE-NL | $12.4_{\pm0.0}$ | $\mathbf{12.4_{\pm0.0}}$ | $20.2_{\pm1.7}$ | 33.3 | $44.3_{\pm1.2}$ | 45.4 |
| DE-PT | $1.6_{\pm0.0}$ | $\mathbf{1.6_{\pm0.0}}$ | $3.3_{\pm0.2}$ | 4.7 | $6.1_{\pm0.1}$ | 6.2 |
| DE-RU | $2.4_{\pm1.4}$ | $15.6_{\pm0.8}$ | $\mathbf{8.4_{\pm3.0}}$ | 25.2 | $39.0_{\pm3.9}$ | 51.2 |
| EN-ES | $62.8_{\pm0.2}$ | $\mathbf{62.6_{\pm0.1}}$ | $113.6_{\pm7.6}$ | 180.0 | $239.4_{\pm7.5}$ | 245.1 |
| EN-FI | $2.6_{\pm0.5}$ | $\mathbf{3.4_{\pm0.0}}$ | $5.7_{\pm0.2}$ | 6.1 | $12.2_{\pm0.6}$ | 12.6 |
| EN-FR | $66.8_{\pm0.1}$ | $\mathbf{66.5_{\pm0.3}}$ | $92.0_{\pm20.2}$ | 233.1 | $260.1_{\pm21.4}$ | 314.0 |
| EN-IT | $22.4_{\pm0.0}$ | $\mathbf{22.3_{\pm0.0}}$ | $39.3_{\pm2.9}$ | 65.1 | $87.0_{\pm2.8}$ | 89.5 |
| EN-NL | $34.2_{\pm0.1}$ | $\mathbf{34.1_{\pm0.0}}$ | $54.3_{\pm3.4}$ | 84.1 | $110.3_{\pm2.1}$ | 111.7 |
| EN-NO | $3.0_{\pm0.0}$ | $\mathbf{2.9_{\pm0.0}}$ | $3.6_{\pm0.9}$ | 8.6 | $8.4_{\pm0.9}$ | 11.8 |
| EN-PL | $1.9_{\pm0.3}$ | $\mathbf{2.2_{\pm0.0}}$ | $4.1_{\pm0.1}$ | 6.4 | $8.7_{\pm0.3}$ | 9.0 |
| EN-PT | $2.2_{\pm0.2}$ | $\mathbf{1.5_{\pm0.0}}$ | $3.1_{\pm0.3}$ | 4.5 | $5.6_{\pm0.2}$ | 5.8 |
| EN-RU | $2.7_{\pm0.9}$ | $13.2_{\pm0.6}$ | $\mathbf{5.5_{\pm2.4}}$ | 23.8 | $30.0_{\pm5.1}$ | 47.8 |
| EN-SV | $3.2_{\pm0.6}$ | $\mathbf{3.3_{\pm0.0}}$ | $4.0_{\pm0.7}$ | 8.6 | $8.7_{\pm0.9}$ | 11.5 |
| ES-FI | $3.4_{\pm0.5}$ | $\mathbf{3.6_{\pm0.0}}$ | $6.2_{\pm0.6}$ | 9.1 | $11.9_{\pm0.3}$ | 12.2 |
| ES-FR | $32.3_{\pm0.1}$ | $\mathbf{32.2_{\pm0.1}}$ | $53.6_{\pm4.8}$ | 103.1 | $116.7_{\pm7.8}$ | 144.0 |
| ES-IT | $24.0_{\pm0.0}$ | $\mathbf{23.9_{\pm0.0}}$ | $41.5_{\pm2.9}$ | 62.6 | $83.4_{\pm2.0}$ | 85.0 |
| ES-NL | $28.7_{\pm0.0}$ | $\mathbf{28.7_{\pm0.0}}$ | $45.7_{\pm3.4}$ | 73.2 | $96.5_{\pm2.1}$ | 97.8 |
| ES-NO | $3.4_{\pm0.0}$ | $\mathbf{3.4_{\pm0.0}}$ | $3.9_{\pm0.7}$ | 9.1 | $8.8_{\pm0.9}$ | 12.4 |
| ES-PT | $1.9_{\pm0.0}$ | $\mathbf{1.5_{\pm0.0}}$ | $3.0_{\pm0.2}$ | 4.5 | $6.2_{\pm0.1}$ | 6.3 |
| ES-RU | $3.6_{\pm1.2}$ | $15.7_{\pm0.0}$ | $\mathbf{3.7_{\pm2.2}}$ | 25.9 | $36.3_{\pm4.5}$ | 50.5 |
| FI-FR | $3.6_{\pm0.3}$ | $\mathbf{6.0_{\pm0.3}}$ | $10.0_{\pm0.3}$ | 11.2 | $10.9_{\pm0.4}$ | 11.2 |
| FI-NO | $4.8_{\pm0.3}$ | $\mathbf{3.5_{\pm0.0}}$ | $6.1_{\pm0.5}$ | 9.1 | $11.4_{\pm0.5}$ | 12.1 |
| FI-PL | $2.8_{\pm0.0}$ | $\mathbf{2.7_{\pm0.0}}$ | $4.8_{\pm0.1}$ | 7.8 | $10.3_{\pm0.4}$ | 10.6 |
| FR-IT | $11.6_{\pm0.0}$ | $\mathbf{11.5_{\pm0.0}}$ | $20.5_{\pm1.4}$ | 32.6 | $43.0_{\pm0.9}$ | 43.7 |
| FR-NL | $30.5_{\pm0.0}$ | $\mathbf{30.5_{\pm0.0}}$ | $49.2_{\pm1.0}$ | 82.9 | $111.5_{\pm3.9}$ | 114.8 |
| FR-NO | $3.1_{\pm0.4}$ | $\mathbf{3.3_{\pm0.0}}$ | $5.3_{\pm0.4}$ | 8.6 | $10.5_{\pm0.4}$ | 11.4 |
| FR-PL | $2.4_{\pm0.3}$ | $\mathbf{2.6_{\pm0.0}}$ | $4.2_{\pm1.0}$ | 6.8 | $9.3_{\pm0.3}$ | 9.5 |
| FR-PT | $1.5_{\pm0.0}$ | $\mathbf{1.5_{\pm0.0}}$ | $2.9_{\pm0.1}$ | 4.7 | $6.2_{\pm0.1}$ | 6.4 |
| FR-RU | $1.6_{\pm0.5}$ | $7.5_{\pm0.4}$ | $\mathbf{3.9_{\pm0.7}}$ | 12.5 | $17.8_{\pm2.3}$ | 25.0 |
| FR-SV | $2.9_{\pm0.4}$ | $\mathbf{3.1_{\pm0.0}}$ | $3.8_{\pm1.2}$ | 8.5 | $8.4_{\pm0.6}$ | 11.4 |
| IT-NL | $2.3_{\pm0.0}$ | $\mathbf{2.3_{\pm0.0}}$ | $4.2_{\pm0.2}$ | 6.2 | $8.7_{\pm0.2}$ | 8.9 |
| IT-PT | $1.8_{\pm0.0}$ | $\mathbf{1.4_{\pm0.0}}$ | $2.9_{\pm0.2}$ | 4.7 | $5.8_{\pm0.1}$ | 5.9 |
| IT-RU | $4.2_{\pm0.0}$ | $16.2_{\pm0.0}$ | $\mathbf{4.3_{\pm1.6}}$ | 27.2 | $37.6_{\pm5.1}$ | 54.4 |
| IT-SV | $3.3_{\pm0.2}$ | $\mathbf{3.3_{\pm0.0}}$ | $4.7_{\pm0.9}$ | 8.7 | $9.1_{\pm0.9}$ | 11.5 |

# C HYPERPARAMETERS

## C.1 FINE-TUNING

We used the Trainer module from Hugging Face's **transformers** library for fine-tuning. The key hyperparameters and settings were as follows:

- Optimizer: AdamW

- Learning rate: $7 \times 10^{-6}$

- Batch size: Automatically determined based on available hardware

- Early stopping patience: 3 epochs

- Mixed precision: Enabled (BF16)

All unspecified settings followed the default values defined in Trainer module.

Table 2: Time in GPU-hours to recover 99.9% of the optimal hypervolume on held-out datasets, optimizing for three objectives: model size, evaluation loss and BLEU score.

| Dataset | Oracle | LAMPS (ours) | MO-ASHA | ZigZag | Blind | Exhaustive |
|---|---|---|---|---|---|---|
| DE-EN | $63.0_{\pm3.7}$ | $88.9_{\pm1.1}$ | $94.5_{\pm4.5}$ | **88.4** | $120.6_{\pm3.3}$ | 123.9 |
| DE-ES | $30.6_{\pm5.1}$ | $51.4_{\pm0.6}$ | $\mathbf{49.0_{\pm6.9}}$ | 53.1 | $71.4_{\pm2.6}$ | 73.6 |
| DE-FR | $26.0_{\pm2.7}$ | $62.7_{\pm3.2}$ | $\mathbf{34.5_{\pm5.6}}$ | 66.1 | $90.4_{\pm2.8}$ | 93.0 |
| DE-IT | $33.9_{\pm1.9}$ | $\mathbf{51.8_{\pm1.1}}$ | $54.6_{\pm13.3}$ | 57.7 | $76.5_{\pm0.9}$ | 77.1 |
| DE-NL | $29.0_{\pm2.5}$ | $\mathbf{32.1_{\pm0.6}}$ | $44.0_{\pm0.8}$ | 45.3 | $44.9_{\pm0.7}$ | 45.4 |
| DE-PT | $3.7_{\pm0.1}$ | $\mathbf{4.2_{\pm0.1}}$ | $6.2_{\pm0.0}$ | 6.2 | $6.2_{\pm0.1}$ | 6.2 |
| DE-RU | $13.3_{\pm1.4}$ | $36.1_{\pm0.1}$ | $\mathbf{17.3_{\pm2.4}}$ | 29.8 | $49.9_{\pm1.4}$ | 51.2 |
| EN-ES | $62.9_{\pm0.1}$ | $\mathbf{94.6_{\pm35.2}}$ | $99.3_{\pm8.8}$ | 180.1 | $241.7_{\pm4.2}$ | 245.1 |
| EN-FI | $7.3_{\pm0.3}$ | $\mathbf{8.3_{\pm0.1}}$ | $11.9_{\pm0.2}$ | 12.6 | $12.2_{\pm0.3}$ | 12.6 |
| EN-FR | $77.6_{\pm3.1}$ | $222.4_{\pm0.0}$ | $\mathbf{126.5_{\pm8.9}}$ | 240.1 | $309.1_{\pm6.3}$ | 314.0 |
| EN-IT | $23.9_{\pm0.0}$ | $62.7_{\pm2.5}$ | $\mathbf{39.5_{\pm2.2}}$ | 65.1 | $87.0_{\pm2.5}$ | 89.5 |
| EN-NL | $61.4_{\pm3.0}$ | $\mathbf{81.9_{\pm1.3}}$ | $104.0_{\pm4.7}$ | 84.1 | $109.6_{\pm2.2}$ | 111.7 |
| EN-NO | $5.8_{\pm0.4}$ | $\mathbf{8.9_{\pm0.5}}$ | $11.2_{\pm0.1}$ | 11.8 | $11.5_{\pm0.3}$ | 11.8 |
| EN-PL | $4.7_{\pm0.0}$ | $\mathbf{5.8_{\pm0.4}}$ | $8.7_{\pm0.3}$ | 9.0 | $8.7_{\pm0.3}$ | 9.0 |
| EN-PT | $3.2_{\pm0.0}$ | $\mathbf{3.6_{\pm0.0}}$ | $5.3_{\pm1.0}$ | 4.5 | $5.7_{\pm0.1}$ | 5.8 |
| EN-RU | $10.6_{\pm0.1}$ | $28.4_{\pm7.4}$ | $\mathbf{17.7_{\pm2.9}}$ | 23.8 | $46.4_{\pm1.5}$ | 47.8 |
| EN-SV | $5.2_{\pm0.3}$ | $\mathbf{7.7_{\pm0.0}}$ | $11.0_{\pm0.1}$ | 11.5 | $10.2_{\pm0.6}$ | 11.5 |
| ES-FI | $6.8_{\pm0.8}$ | $\mathbf{8.8_{\pm0.3}}$ | $11.6_{\pm0.0}$ | 12.2 | $11.6_{\pm0.4}$ | 12.2 |
| ES-FR | $86.5_{\pm1.2}$ | $\mathbf{99.4_{\pm2.0}}$ | $143.8_{\pm0.0}$ | 143.7 | $142.6_{\pm2.2}$ | 144.0 |
| ES-IT | $43.6_{\pm0.7}$ | $\mathbf{59.1_{\pm1.8}}$ | $81.8_{\pm1.6}$ | 64.3 | $84.4_{\pm0.9}$ | 85.0 |
| ES-NL | $66.5_{\pm2.2}$ | $\mathbf{81.9_{\pm4.9}}$ | $97.7_{\pm0.0}$ | 97.7 | $96.9_{\pm1.3}$ | 97.8 |
| ES-NO | $6.7_{\pm0.7}$ | $\mathbf{7.5_{\pm0.3}}$ | $12.4_{\pm0.0}$ | 12.4 | $11.9_{\pm0.5}$ | 12.4 |
| ES-PT | $3.9_{\pm0.1}$ | $\mathbf{3.8_{\pm0.2}}$ | $6.3_{\pm0.0}$ | 6.3 | $6.3_{\pm0.1}$ | 6.3 |
| ES-RU | $12.5_{\pm3.4}$ | $34.5_{\pm1.1}$ | $\mathbf{19.0_{\pm0.9}}$ | 25.9 | $48.8_{\pm1.4}$ | 50.5 |
| FI-FR | $5.1_{\pm0.0}$ | $\mathbf{7.0_{\pm0.5}}$ | $10.6_{\pm0.1}$ | 11.2 | $10.9_{\pm0.3}$ | 11.2 |
| FI-NO | $8.6_{\pm0.3}$ | $\mathbf{8.4_{\pm0.1}}$ | $12.1_{\pm0.0}$ | 12.1 | $11.9_{\pm0.3}$ | 12.1 |
| FI-PL | $6.6_{\pm0.2}$ | $\mathbf{7.1_{\pm0.4}}$ | $10.6_{\pm0.0}$ | 10.6 | $10.3_{\pm0.4}$ | 10.6 |
| FR-IT | $18.7_{\pm0.0}$ | $30.5_{\pm0.3}$ | $\mathbf{20.1_{\pm1.5}}$ | 32.6 | $42.9_{\pm1.1}$ | 43.7 |
| FR-NL | $74.3_{\pm5.2}$ | $\mathbf{80.2_{\pm1.0}}$ | $114.8_{\pm0.0}$ | 114.6 | $113.9_{\pm1.6}$ | 114.8 |
| FR-NO | $6.2_{\pm0.7}$ | $\mathbf{6.7_{\pm0.4}}$ | $11.4_{\pm0.0}$ | 11.4 | $11.2_{\pm0.2}$ | 11.4 |
| FR-PL | $5.6_{\pm0.2}$ | $\mathbf{5.9_{\pm0.0}}$ | $9.3_{\pm0.2}$ | 9.5 | $9.3_{\pm0.2}$ | 9.5 |
| FR-PT | $3.6_{\pm0.2}$ | $\mathbf{4.3_{\pm0.1}}$ | $6.4_{\pm0.0}$ | 6.4 | $6.3_{\pm0.1}$ | 6.4 |
| FR-RU | $7.3_{\pm0.1}$ | $13.3_{\pm3.1}$ | $\mathbf{10.8_{\pm0.2}}$ | 18.6 | $24.5_{\pm0.7}$ | 25.0 |
| FR-SV | $7.2_{\pm0.4}$ | $\mathbf{7.7_{\pm0.1}}$ | $11.4_{\pm0.0}$ | 11.4 | $11.1_{\pm0.3}$ | 11.4 |
| IT-NL | $6.1_{\pm0.2}$ | $\mathbf{6.6_{\pm0.5}}$ | $8.9_{\pm0.0}$ | 8.9 | $8.8_{\pm0.1}$ | 8.9 |
| IT-PT | $3.8_{\pm0.1}$ | $\mathbf{4.2_{\pm0.1}}$ | $5.7_{\pm0.3}$ | 5.9 | $5.9_{\pm0.0}$ | 5.9 |
| IT-RU | $13.0_{\pm1.5}$ | $35.3_{\pm1.8}$ | $\mathbf{11.6_{\pm2.7}}$ | 27.2 | $50.6_{\pm2.6}$ | 54.4 |
| IT-SV | $6.9_{\pm0.4}$ | $\mathbf{7.6_{\pm0.1}}$ | $11.5_{\pm0.0}$ | 11.5 | $11.2_{\pm0.4}$ | 11.5 |

## C.2 PPO

For the PPO algorithm, we used the implementation from Stable Baselines3 library. The key hyper-parameters and settings were as follows:

- Learning rate: $1 \times 10^{-4}$
- Minibatch size: 256
- Num. epochs: 15
- Discount ($\gamma$): 0.99
- GAE parameter ($\lambda$): 0.97
- Clip range: 0.20
- VF coeff. $c_1$: 0.5
- Entropy coeff. $c_2$: 0.23

All policies were trained using the Gymnasium environment API with invalid action masking.

## D    EMPIRICAL CONVERGENCE ANALYSIS OF THE REWARDS

In order to provide additional evidence that the reward function defined in equation 7 effectively guides the agent toward the optimal solution set, according to the original multi-objective problem in equation 1, Figure 5 presents a typical reward evolution observed during training, for both single task and multi-task RL (MTRL) using PPO .

For better interpretability and comparison, reward values are normalized such that a value of 3000 corresponds to the optimal reward, when the agent exclusively evaluates Pareto-optimal solutions, achieving maximal hypervolume in minimal time. The learned policy exhibits a consistent upward trend in reward, eventually converging to the optimal value.

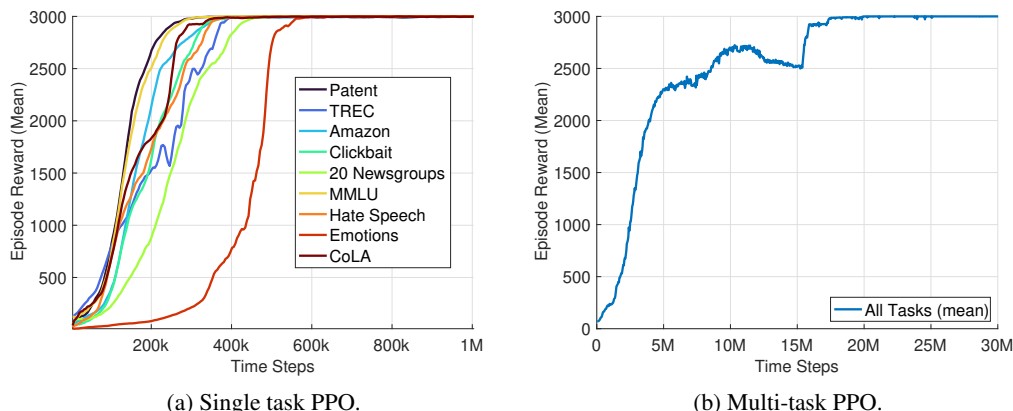

(a) Single task PPO.                            (b) Multi-task PPO.

Figure 5: Normalized reward progression during policy training using PPO algorithm. As expected, multi-task RL takes longer, but it also converges to the optimal reward.

## E    DATASETS

This section describes the datasets used in our experiments.

### E.1    TEXT CLASSIFICATION

Although the datasets described here correspond to text classification tasks, they cover different NLP tasks, requiring different linguistic competencies, domain knowledge, and reasoning abilities. This diversity makes it particularly challenging (and well-suited) for evaluating LAMPS. For datasets without predefined training and validation splits, we reserve 20% of the data for validation.

**TREC**    A classic question classification benchmark with 6 coarse-grained classes (e.g., abbreviation, entity, description and abstract concept, human being, location, and numeric value). Task: Question classification. License: N/A (widely used academic benchmark; originally from UIUC).

**Clickbait**    Contains news headlines labeled as either "clickbait" or "non-clickbait". Derived from social media posts (Chakraborty et al., 2016). Task: Binary classification. License: N/A.

**20 Newsgroups**    A collection of $20,000$ newsgroup emails across 20 different topics (Lang, 1995). Task: Topic classification. License: CC BY 4.0.

**Amazon Reviews (cell-phone)**    Subset of the Amazon Product Review 2013 dataset, filtered for the "Cell Phone reviews" category. Includes star ratings from 1 to 5 and contains $78,930$ reviews. Task: Sentiment classification (5 classes). License: N/A (Amazon public data, widely used in academia).

**Hate Speech and Offensive Language**   A corpus of over 24,000 tweets manually annotated as hate speech, offensive but not hateful, or neither (Davidson et al., 2017). Task: Offensive language classification (3 classes). License: MIT License.

**MMLU**   Massive Multitask Language Understanding, a benchmark covering 57 diverse subject areas from elementary math to law and philosophy. Task: Multi-choice question answering. License: MIT License.

**Patent Classification**   Consisting of $35,000$ Patent abstracts labeled with Cooperative Patent Classification (CPC) codes (9 classes). Task: Topic classification. License: Public domain (based on USPTO data).

**Emotion**   A dataset of $20,000$ Twitter messages in English annotated with one of six basic emotions (anger, fear, joy, love, sadness, surprise). Task: Emotion classification. License: MIT License.

**CoLA**   Corpus of Linguistic Acceptability, a dataset of English sentences labeled as grammatically acceptable or unacceptable. Task: Acceptability classification (binary). License: Unknown (academic benchmark from the GLUE suite).

**SMS Spam**   A dataset of SMS messages labeled as spam or ham, widely used in spam detection research. Task: Binary classification. License: Open for research use.

**IMDB**   A large-scale movie review corpus containing 50K reviews labeled as positive or negative (Maas et al., 2011). Task: Sentiment classification (binary). License: Permissive research license.

**Financial Phrasebank**   A financial-domain sentiment dataset of short sentences annotated by multiple experts with high-agreement labels (positive, negative, neutral) (Malo et al., 2014). Task: Financial sentiment analysis (3 classes). License: Creative Commons Attribution-NonCommercial-ShareAlike 3.0 Unported License.

### E.2    MACHINE TRANSLATION

**OPUS Books**   A collection of copyright-free texts translated into multiple languages (Tiedemann, 2012). License: Available for personal, educational and research use.

## F    PRETRAINED LANGUAGE MODELS

Below is the list of pretrained models used during the experiments of this paper:

### F.1    TEXT CLASSIFICATION

**BERT:**

1. google-bert/bert-large-cased-whole-word-masking
2. google-bert/bert-large-uncased-whole-word-masking-fine-tuned-squad
3. google-bert/bert-large-uncased-whole-word-masking
4. google-bert/bert-large-uncased
5. google-bert/bert-large-cased-whole-word-masking-fine-tuned-squad
6. google-bert/bert-large-cased
7. google-bert/bert-base-uncased
8. google-bert/bert-base-multilingual-uncased
9. google-bert/bert-base-multilingual-cased
10. google-bert/bert-base-german-dbmdz-uncased
11. google-bert/bert-base-german-dbmdz-cased
12. google-bert/bert-base-german-cased
13. google-bert/bert-base-chinese
14. google-bert/bert-base-cased

**GPT:**

1. openai-community/gpt2
2. openai-community/gpt2-medium
3. openai-community/gpt2-large
4. openai-community/gpt2-xl

**RoBERTa:**

1. FacebookAI/roberta-base
2. FacebookAI/roberta-large
3. FacebookAI/xlm-roberta-base
4. FacebookAI/xlm-roberta-large
5. FacebookAI/xlm-roberta-large-fine-tuned-conll02-dutch
6. FacebookAI/xlm-roberta-large-fine-tuned-conll02-spanish
7. FacebookAI/xlm-roberta-large-fine-tuned-conll03-english
8. FacebookAI/xlm-roberta-large-fine-tuned-conll03-german

**OPT:**

1. facebook/opt-125m
2. facebook/opt-350m
3. facebook/opt-1.3b
4. facebook/opt-2.7b
5. facebook/opt-6.7b

**Llama:**

1. meta-llama/Llama-3.2-1B
2. meta-llama/Llama-3.2-1B-Instruct
3. meta-llama/Llama-3.2-3B
4. meta-llama/Llama-3.1-8B

**DistilBERT:**

1. distilbert/distilbert-base-multilingual-cased
2. distilbert/distilbert-base-german-cased
3. distilbert/distilbert-base-uncased-distilled-squad
4. distilbert/distilbert-base-cased-distilled-squad
5. distilbert/distilbert-base-cased
6. distilbert/distilbert-base-uncased
7. distilbert/distilroberta-base
8. distilbert/distilgpt2

**ALBERT:**

1. albert/albert-xlarge-v2
2. albert/albert-xxlarge-v2
3. albert/albert-xxlarge-v1
4. albert/albert-xlarge-v1
5. albert/albert-large-v2
6. albert/albert-large-v1
7. albert/albert-base-v2
8. albert/albert-base-v1

**LUKE:**

1. studio-ousia/mluke-large
2. studio-ousia/mluke-large-lite

3. studio-ousia/mluke-base-lite
4. studio-ousia/mluke-base
5. studio-ousia/luke-japanese-base
6. studio-ousia/luke-japanese-base-lite
7. studio-ousia/luke-japanese-large-lite
8. studio-ousia/luke-japanese-large
9. studio-ousia/luke-large-lite
10. studio-ousia/luke-base-lite
11. studio-ousia/luke-large
12. studio-ousia/luke-base

**DeepSeek:**

1. deepseek-ai/DeepSeek-R1-Distill-Qwen-1.5B
2. deepseek-ai/DeepSeek-R1-Distill-Qwen-7B
3. deepseek-ai/DeepSeek-R1-Distill-Llama-8B

**Qwen:**

1. Qwen/Qwen2.5-0.5B
2. Qwen/Qwen2.5-1.5B
3. Qwen/Qwen2.5-3B
4. Qwen/Qwen2.5-7B

## F.2 MACHINE TRANSLATION

**Helsinki-NLP:**

1. Helsinki-NLP/opus-mt-en-sv
2. Helsinki-NLP/opus-mt-tc-bible-big-deu_eng_fra_por_spa-mul

**mBART:**

1. facebook/mbart-large-50
2. facebook/mbart-large-50-many-to-many-mmt
3. facebook/mbart-large-50-many-to-one-mmt
4. facebook/mbart-large-50-one-to-many-mmt
5. facebook/mbart-large-cc25
6. facebook/mbart-large-en-ro

**T5:**

1. google-t5/t5-3b
2. google-t5/t5-base
3. google-t5/t5-large
4. google-t5/t5-small
5. google/long-t5-local-large
6. google/long-t5-tglobal-xl

**mT5:**

1. google/mt5-base
2. google/mt5-large
3. google/mt5-small
4. google/mt5-xl

**UMT5:**

1. google/umt5-base
2. google/umt5-small

## G  ADDING NEW MODELS TO THE META-DATASET

To incorporate a new model into the recommendation pool of LAMPS, it must first be integrated into the meta-dataset. We refer to this process as *model fingerprinting*. Because LAMPS relies on meta-learning, it is necessary to observe the actual performance of the new model on known datasets before the system can generalize its behavior to unseen datasets. This integration requires two steps:

1. The new LLM must be fine-tuned on all datasets currently included in the meta-dataset, with all relevant metrics recorded.

2. The reinforcement learning policy must be retrained on the expanded meta-dataset.

Currently, complete retraining is the recommended procedure for reliable integration of new models. Although incremental training strategies could further reduce the computational overhead, the cost of full retraining is already negligible compared to the fine-tuning runs required to expand the meta-dataset.

The ideal number and diversity of datasets in the meta-dataset remains an open research question. A smaller set of datasets facilitates the addition of new models, since each integration requires fewer fine-tuning runs. Conversely, a larger and more diverse collection typically improves the generalization ability of the learned policy to unseen tasks. How to balance these competing goals remains an open challenge for future work.

## H  PROOF OF THEOREM 1

*Proof.* We first prove that the maximizer $X_\lambda = \arg\max_{X \subset \mathcal{X}} H_\mathcal{D}(X, r) - \lambda|X|$ is a subset of Pareto solutions $X^*$, that is, for any $\lambda > 0$, $X_\lambda \subset X^*$. This is proved by contradiction. Suppose that there exists a $x \in X_\lambda$ that is not Pareto-optimal. Then, there exists a $x^* \in \mathcal{X}$ dominating $x$ such that $\Lambda([x, r]) < \Lambda([x^*, r])$ holds. Denote $X_\lambda^*$ the set obtained from $X_\lambda$ by replacing $x$ with $x^*$. By the definition of $H_\mathcal{D}$, we know $H_\mathcal{D}(X_\lambda, r) < H_\mathcal{D}(X_\lambda^*, r)$. Then, it holds

$$H_\mathcal{D}(X_\lambda, r) - \lambda|X_\lambda| = H_\mathcal{D}(X_\lambda, r) - \lambda|X_\lambda^*| < H_\mathcal{D}(X_\lambda^*, r) - \lambda|X_\lambda^*|.$$

This contradicts the assumption that $X_\lambda$ is the maximizer of problem (3). Hence, for any $\lambda > 0$, we know $X_\lambda \subset X^*$. Below we prove the *if* part and the *only if* part respectively. **The *if* part:** In this part, we prove that if equation 4 holds, then the optimal solution $X^* \subset \mathcal{X}$ of problem equation 3 contains only and exactly all Pareto-optimal solutions. Let $X_\lambda = \arg\max_{X \subset \mathcal{X}} H_\mathcal{D}(X, r) - \lambda|X|$. From the above discussion we know $X_\lambda \subset X^*$. Suppose $|X^*| - |X_\lambda| = s$. We denote $\{x_{i_1}, \ldots, x_{i_s}\} \subset X^*$ the subset of $X^*$ such that $\{x_{i_1}, \ldots, x_{i_s}\} \cap X_\lambda = \varnothing$. We define $X_k = X_\lambda \cup \{x_{i_1}, \ldots, x_{i_k}\}$ for all $k \in \{0, 1, \ldots, s\}$. Then, we know $X_s = X^*$, $X_0 = X_\lambda$ and $|X_{k+1}| - |X_k| = 1$ for all $k \in \{0, 1 \ldots, s-1\}$. Note that

$$
\begin{aligned}
H_\mathcal{D}(X^*, r) &- \lambda|X^*| - \big(H_\mathcal{D}(X_\lambda, r) - \lambda|X_\lambda|\big) \\
&= H_\mathcal{D}(X^*, r) - H_\mathcal{D}(X_\lambda, r) - \lambda\big(|X^*| - |X_\lambda|\big) \\
&= H_\mathcal{D}(X_s, r) - H_\mathcal{D}(X_0, r) - s\lambda \\
&= \sum_{k=1}^{s} \big(H_\mathcal{D}(X_k, r) - H_\mathcal{D}(X_{k-1}, r) - \lambda\big) \\
&\geq \sum_{k=1}^{s} \Big(H_\mathcal{D}(X_k, r) - H_\mathcal{D}(X_{k-1}, r) - \min_{x \in X, X \subset X^*} \Delta H_\mathcal{D}(x|X)\Big) \\
&\geq 0,
\end{aligned}
\tag{8}
$$

where the last second inequality used equation 4 and the last inequality used the definition of $\min_{x \in X, X \subset X^*} \Delta H_\mathcal{D}(x|X)$. **The *only if* part:** To prove this part of the result, we only need to show that there exists an optimization problem whose Pareto solution set $X^*$ with $|X^*| = s$ satisfies that for any sequence of subsets $\{X_i\}_{i=1}^{s}$ satisfying $X_i \subset X^*$ and $|X_i| = i$, it holds

$$\max_{i \in \{2, \ldots, s\}} H_\mathcal{D}(X_i, r) - H_\mathcal{D}(X_{i-1}, r) = \min_{x \in X, X \subset X^*} \Delta H_\mathcal{D}(x|X). \tag{9}$$

On the other hand, from equation 8 and the definition of $X^*$ we know

$$H_{\mathcal{D}}(X^*, r) - \lambda|X^*| - \left(H_{\mathcal{D}}(X_\lambda, r) - \lambda|X_\lambda|\right) = \sum_{k=1}^{s} \left(H_{\mathcal{D}}(X_k, r) - H_{\mathcal{D}}(X_{k-1}, r)\right) - s\lambda \geq 0.$$

Hence, we have $\sum_{k=1}^{s} \left(H_{\mathcal{D}}(X_k, r) - H_{\mathcal{D}}(X_{k-1}, r)\right) \geq s\lambda$. Combining this observation with equation 9 together, we get

$$\min_{x \in X, X \subset X^*} \Delta H_{\mathcal{D}}(x|X) \geq \lambda.$$

The proof is completed by noting that equation 9 always holds for arbitrary $X^*$ with $|X^*| = 2$. $\quad\square$

## I  THE USE OF LARGE LANGUAGE MODELS (LLMs)

We used large language models solely for surface-level editing: spelling and grammar correction, and minor wording improvements. LLMs were *not* used for idea generation, experiment design, data analysis, coding, mathematical derivations, or substantive content creation.

