# OpenReview forum: "Landmark-Guided Policy Optimization for Multi-Objective Language Model Selection"
_ICLR.cc/2026/Conference — Submitted to ICLR 2026_

### Official Review · Reviewer_4aCY · 2025-10-20

**Soundness:** 3
**Presentation:** 3
**Contribution:** 2
**Rating:** 4
**Confidence:** 3

**Summary:**

LAMPS is a language model selection framework that aims to choose the best model for a particular task. The authors claim that the best model is not necessarily the one with the lowest test loss but could be influenced by many other factors (resources, deployment costs). Hence, in the context of AutoML, finding the best model is expensive. They formulate the problem as an RL problem (trained with PPO) with an objective function of finding a Pareto optimal set over the models, by minimizing the joint cost for all factors (model size, inference throughput, deployment cost) (defined as the hypervolume indicator).

**Strengths:**

- The paper has well formatted definitions and theorems
- The figures are well made
- The problem they choose to solve is interesting.

**Weaknesses:**

- The related works section could be strengthened. From what I remember, there are works that try to automate neural architecture search, which seems like a similar problem setting. I understand that experiments would be too much to run, but at least including a discussion on whether these methods could be adapted to this task/setting would make the claims stronger.
- It can be expensive to run all the configurations before deciding what model to select.

**Questions:**

1. Is my understanding correct: the multi-objective RL comes in from the fact that each factor (test loss, model size, inference throughput, deployment cost) is a dimension in the hypervolume indicator ($i=0$ for test, $i=1$ for model size, etc.), and these factors are all aggregated with the Lebesgue measure?
2. Is there any analysis on how the conflicting factors affect the performance of LAMPS? What if there were no conflicting factors? What if there were only conflicting factors?

---

> ### Author Response · Authors · 2025-11-19
>
> We thank the reviewer for taking the time to provide comments on our work. We appreciate the opportunity to clarify some points raised and to provide additional details that may help re-evaluate our work.
>
> > The related works section could be strengthened. From what I remember, there are works that try to automate neural architecture search, which seems like a similar problem setting. I understand that experiments would be too much to run, but at least including a discussion on whether these methods could be adapted to this task/setting would make the claims stronger.
>
> Please, check our general response where we compare LAMPS with ASHA+EpsNet, a multi-objective method used in NAS. Thank you for the suggestion.
>
> > It can be expensive to run all the configurations before deciding what model to select.
>
> There is a misunderstanding here. LAMPS does not run all configurations, but quite the opposite: it is specifically designed to evaluate as few configurations (models) as possible. As shown in Figure 3, the `Exhaustive` baseline corresponds to evaluating all models, and it is clear that LAMPS dramatically reduces the search time while remaining comparable to the `Oracle` in 7 out of 9 held-out datasets. Similar results were also obtained for the new experiments on machine-translation.
>
> > Is my understanding correct: the multi-objective RL comes in from the fact that each factor (test loss, model size, inference throughput, deployment cost) is a dimension in the hypervolume indicator (i=0 for test, i=1 for model size, etc.), and these factors are all aggregated with the Lebesgue measure?
>
> Yes, this understanding is correct. This is illustrated in Figure 1, where the Lebesgue measure collapses to the area formed by the solution set and the reference point.
>
> > Is there any analysis on how the conflicting factors affect the performance of LAMPS? What if there were no conflicting factors? What if there were only conflicting factors?
>
> Because LAMPS is a multi-objective framework, it is most beneficial when the objectives are conflicting. If the objectives are highly correlated, the search effectively becomes one-dimensional and the Pareto front collapses to a single dominant solution (e.g., the model with the best performance). Although LAMPS can still operate in that regime, it is not its primary purpose.

---

> > ### Comment · Reviewer_4aCY · 2025-11-23
> >
> > Thank you for the clarification -- I'll raise my score in the hope that you address some of these comments in the final version of your paper.

---

### Official Review · Reviewer_8m2N · 2025-10-30

**Soundness:** 2
**Presentation:** 2
**Contribution:** 4
**Rating:** 4
**Confidence:** 3

**Summary:**

This paper presents LAMPS, a novel multi-objective AutoML framework for selecting pretrained large language models (LLMs). The authors treat the practical challenge as a multi-objective problem, balancing the trade off between model performance, training cost, etd. LAMPS combines landmark fine-tuning, meta-learning via reinforcement learning, which trains a policy on historical model performance. In the experimental results, LAMPS reduces search time by 71% while maintaining coverage of over 98% of the optimal target-space hypervolume.

**Strengths:**

The paper Introduces a novel method, LAMPS within a multi-objective AutoML framework for efficiently selecting and fine-tuning models along a Pareto front. It combines multi-objective optimization with invalid action masking in RL, which is a novel way to improve exploration efficiency and reduce wasted computation. LAMPS consistently identifies near-Pareto-optimal models faster than baselines

**Weaknesses:**

The termination condition assumes the agent can detect when all Pareto-optimal models have been fully fine-tuned, which is not practical for the real scenario.

The RL agent training rely on fully fine tuning trajectories of 70 pretrained models, which limits its applicability to scenarios involving a large number of pretrained models.

The experiments use only nine datasets and two objectives, more experiments would strengthen the generality and demonstrate its performance on more objective trade-offs. It’s unclear how LAMPS scales when more objectives are added.

**Questions:**

The paper does not clearly define what constitutes a *fully fine-tuned* mode, could the authors specify the exact convergence condition?

The paper uses a sparse reward defined only at the end of each episode (t = T). Have the authors considered using intermediate rewards at each timestep t, for example, based on the hypervolume of the partially fine-tuned model’s current performance?

---

> ### Author Response · Authors · 2025-11-19
>
> We thank the reviewer for the thoughtful and constructive feedback. We appreciate the opportunity to clarify several aspects of our method and address the concerns raised.
>
> > The termination condition assumes the agent can detect when all Pareto-optimal models have been fully fine-tuned, which is not practical for the real scenario.
>
> Thank you for the opportunity to clarify this point. The termination condition that assumes knowledge of when all Pareto-optimal models have been fully fine-tuned is used only during policy training, where privileged information is available. At test time on new datasets, the agent has no access to this information, neither need it: it simply observes the evolving learning curves, selects a model to fine-tune for one additional epoch, and continues until the user-defined search budget is exhausted (Algorithm 1, line 4). Thus, the method does not rely on impractical assumptions in real-world deployment.
>
> > The paper does not clearly define what constitutes a fully fine-tuned mode, could the authors specify the exact convergence condition?
>
> A model is considered fully fine-tuned when its validation loss stops improving for a fixed number of consecutive epochs. In our experiments, we use an early-stopping patience of 3 epochs: if the loss does not improve for 3 consecutive epochs, training is halted and the model is treated as fully fine-tuned, with the best-performing checkpoint retained. We will clarify this definition in the revised version.
>
> > The paper uses a sparse reward defined only at the end of each episode (t = T). Have the authors considered using intermediate rewards at each timestep t, for example, based on the hypervolume of the partially fine-tuned model’s current performance?
>
> We chose to keep the sparse reward for three main reasons. First, switching to a dense reward would break the theoretical foundations established in Theorem 1. Second, dense rewards would require computing hypervolume increments at every timestep, which is computationally expensive and would substantially slow down policy training. Third, the current sparse reward already delivers strong performance. Given these considerations, we do not see sufficient motivation to explore dense rewards in this work.
>
> ---
>
> We hope these clarifications, together with our general response, help address most of the reviewer's concerns and contribute to a positive reassessment of our work.

---

### Official Review · Reviewer_jMZd · 2025-10-30

**Soundness:** 2
**Presentation:** 3
**Contribution:** 3
**Rating:** 4
**Confidence:** 3

**Summary:**

The paper addresses the problem of selecting and fine-tuning a subset of pretrained LLMs to approximate the Pareto front across multiple, potentially conflicting objectives, such as test loss and fine-tuning time. The authors formulate this as a hypervolume maximization problem with a cardinality regularization term, and establish a sufficient condition under which the optimizer recovers the true Pareto set.
Their proposed method, LAMPS, uses early segments of model learning curves, termed "landmarks," to predict final performance. These landmarks guide a reinforcement learning allocation policy, trained with Distral-regularized PPO, which decides which model to allocate further fine-tuning resources to. The policy receives a sparse reward that encourages efficient recovery of the full Pareto set, and incorporates mechanisms like invalid action masking to avoid unnecessary computation.
The deployment procedure begins with a zero-shot evaluation of all candidate models, followed by an iterative process of fine-tuning and evaluation under a budget constraint. The final output consists of non-dominated solutions. The method is meta-trained on a large dataset of full fine-tuning trajectories covering 70 models and nine classification datasets. Experiments show that LAMPS reaches 98 percent of the optimal hypervolume more quickly than baseline methods, achieving performance comparable to an oracle on most tasks while reducing wall-clock search time by 71 percent on average.

Using hypervolume to evaluate Pareto sets is standard in multi-objective optimization (MOO) and maximizing it aligns with Pareto optimality, but the paper’s contribution is to meta-learn a policy that allocates scarce fine-tuning budget across models to rapidly approach a good Pareto set in LLM selection.

**Strengths:**

- The paper introduces a novel framing of model selection and fine-tuning as a policy learning problem that directly optimizes hypervolume. The sparse reward is well-designed and aligned with the set-level objective.

- The formulation combining hypervolume and ℓ₀-style regularization is mathematically sound, with a clear condition on the regularization parameter. The reinforcement learning setup is thoughtful, using Distral for transfer, PPO for stability, and invalid-action masking to improve efficiency.

- The experiments cover a large and realistic search space with 70 pretrained LLMs across nine datasets. Results are reported with respect to wall-clock time, showing a 71 percent average reduction in search time to reach 98 percent of optimal hypervolume. Comparative performance is clearly illustrated in Figure 3.

Deployment details in Algorithm 1 and the inclusion of example commands and a policy checkpoint improve reproducibility and ease of adoption.

- The focus on hypervolume as a set-level objective better reflects practical needs in multi-objective LLM selection and offers a promising path toward reducing compute costs in real-world settings.

**Weaknesses:**

1.	The initial state and per-step evaluation use D_test, which risks peeking at the test set while adaptively selecting and training models, potentially biasing results. A pure validation set (or cross-validation) should drive decisions; the test set should be reserved for final evaluation only.
2.	Baselines are underspecified relative to the literature. The paper compares against Blind/ZigZag/Oracle but omits strong, well-known multi-fidelity HPO and early-stopping methods (e.g., Hyperband/ASHA, BOHB [7]) and learning-curve extrapolation (Domhan et al.) [2], which directly trade resource vs. accuracy and are relevant to the same compute-constrained setting. It also omits multi-objective HPO baselines (e.g., scalarization approaches like ParEGO [3], hypervolume-based MOBO, or MO-ASHA [4]). Without these, the magnitude of LAMPS’ advantage is harder to assess [1].
3.	Experiments optimize only test cross-entropy and training time for classification tasks. Important deployment objectives (e.g., inference throughput/latency, VRAM, energy, dollar cost, fairness/robustness metrics, factuality metrics, or generation-quality metrics for seq2seq) are untested. The claim of objective-agnosticism would be stronger with such evaluations.
4.	Ablations / design analyses are missing or light:
o	Effect of landmark schedule (number and spacing) on prediction fidelity and policy quality.
o	Reward shaping alternatives and sensitivity to the terminal-only reward.
o	Role of Distral vs. single-task policies.
o	Effect of invalid-action masking removal.
o	Sensitivity to the reference point (r) for hypervolume. (The literature notes hypervolume can be reference-point sensitive.) [5]
5.	Training uses 8× A100-40GB and advises ≥2 TB disk; this may be a barrier for many labs. Reporting total policy training cost and amortization analysis vs. per-dataset gains would help.
6.	Relation to hypervolume-maximizing selection. Since LAMPS ultimately maximizes hypervolume of a set, comparisons or discussion against hypervolume-driven selection algorithms (e.g., SMS-EMOA [6]) would contextualize design choices and computational efficiency.

References:
[1] https://arxiv.org/abs/1603.06560
[2] https://www.ijcai.org/Proceedings/15/Papers/487.pdf
[3] https://ieeexplore.ieee.org/document/1583627
[4] https://arxiv.org/pdf/2106.12639
[5] https://arxiv.org/pdf/2005.00515
[6] https://www.sciencedirect.com/science/article/pii/S0377221706005443
[7] https://arxiv.org/pdf/1807.01774

**Questions:**

1.	Can you re-run the main results with decisions driven exclusively by a validation split (no access to test during policy rollouts) and report test-only results at the end? This would address potential test leakage.
2.	It could be instructive to include:
o	Hyperband/ASHA (single-objective time-aware early-stopping) configured for either joint scalarization (e.g., weighted sum of objectives) or treating training time as the resource.
o	BOHB (combines BO with Hyperband).
o	Learning-curve extrapolation (Domhan et al.) to decide early continuation/termination.
o	Multi-objective HPO like ParEGO (scalarization BO) or MO-ASHA. How does LAMPS compare in wall-clock to 98% hypervolume?
3.	Could you add experiments with (i) generation tasks (e.g., summarization), (ii) inference-time/VRAM/energy objectives, and (iii) robustness/fairness/factuality metrics to demonstrate objective-agnosticism?
4.	It may be useful to provide: (a) landmark schedule sensitivity; (b) reward variants (e.g., per-step dense rewards); (c) with/without Distral; (d) invalid-action masking off; (e) reference-point (r) sensitivity.
5.	What reference point (r) is used and how chosen per dataset? Any normalization across objectives?
6.	What is the total compute to train the 45M-step policy, and how many target datasets does it take to amortize that cost compared to (say) BOHB?
7.	Since some Hugging Face models require license acceptance, how is this handled in automated runs (e.g., CI or cluster) to ensure compliance?

---

> ### Author Response · Authors · 2025-11-19
>
> We thank the reviewer for constructive feedbacks. In the following, we address the main concerns raised.
>
> > Can you re-run the main results with decisions driven exclusively by a validation split (no access to test during policy rollouts) and report test-only results at the end? This would address potential test leakage.
>
> The reviewer's concern stems from a misunderstanding of the role of D_test in Algorithm 1. The dataset D used during policy evaluation is entirely held out from policy training, and its split into D_train and D_test is used solely for the inner fine-tuning loop of candidate LLMs. In this context, our notation was indeed misleading: D_test should have been denoted D_eval, since it functions purely as an evaluation split for early stopping and performance estimation during fine-tuning. We have corrected this in the revised manuscript. No data from policy-evaluation datasets is ever used during policy training, so no leakage is possible. Moreover, since LAMPS avoids fully fine-tuning most candidate models, a true held-out test set (if available) would not influence the search process and would only be used to report the final performance of the recovered Pareto front.
>
> > It could be instructive to include: o Hyperband/ASHA (single-objective time-aware early-stopping) configured for either joint scalarization (e.g., weighted sum of objectives) or treating training time as the resource. o BOHB (combines BO with Hyperband). o Learning-curve extrapolation (Domhan et al.) to decide early continuation/termination. o Multi-objective HPO like ParEGO (scalarization BO) or MO-ASHA. How does LAMPS compare in wall-clock to 98% hypervolume?
>
> Thank you for the suggestion. Please check our general response, where we compare LAMPS with MO-ASHA (the ASHA+EpsNet variation, ref. [4]), a multi-objective method closely aligned with our setting. We also include comparisons with the classic single-objective baselines (Successive Halving, Hyperband, and BOHB) by evaluating LAMPS in a single-objective optimization setting. These additional results are presented and discussed in the general response.
>
> > It may be useful to provide: (a) landmark schedule sensitivity; (b) reward variants (e.g., per-step dense rewards); (c) with/without Distral; (d) invalid-action masking off; (e) reference-point (r) sensitivity.
>
> - b) We chose to keep the sparse reward for three main reasons. First, switching to a dense reward would break the theoretical foundations established in Theorem 1. Second, dense rewards would require computing hypervolume increments at every timestep, which is computationally expensive and would substantially slow down policy training. Third, the current sparse reward already delivers strong performance. Given these considerations, we do not see sufficient motivation to explore dense rewards in this work.
> - c) Initially, we tested LAMPS without Distral, using vanilla multi-task RL (MTRL). In fact, Distral offered only marginal improvements over MTRL. So, we are considering reporting only MTRL instead of Distral, making it easier for others to reproduce our work (and possibly improve it).
> - d) Our initial experiments without invalid-action masking (penalizing invalid actions with a negative reward instead) were consistent with [Teh et al., 2020](https://arxiv.org/pdf/2006.14171): training was longer, less stable, and yielded worse performance.
>
> > What reference point (r) is used and how chosen per dataset? Any normalization across objectives?
>
> The reference point is used only during policy training, being required for computing the hypervolume. For each dataset, we simply choose a reference point that is slightly dominated by all observed points in the objective space (+10% on each objective). Regarding normalization, we normalize the hypervolume by the optimal hypervolume (page 8, line 428), accounting for all objectives.
>
> > What is the total compute to train the 45M-step policy, and how many target datasets does it take to amortize that cost compared to (say) BOHB?
>
> Once a meta-dataset is available, training the policy does not require GPUs and it can be done on a standard consumer-grade machine. For example, on a MacBook Pro M2 (16 GB RAM), training a policy with 8 meta-datasets runs at approximately 5K steps per second, so a 45M-step policy is trained in roughly 2.5 hours. Once a trained policy is available, it is intended to be reused by end users, so the only compute cost for them is the GPU budget available for the search on a new dataset.
>
> > Since some Hugging Face models require license acceptance, how is this handled in automated runs (e.g., CI or cluster) to ensure compliance?
>
> Gated models can be accessed through access tokens, which can be set using environment variables. Once access to a model has been granted for a given token, no further human interaction is required.

---

### Author Response · Authors · 2025-11-19
**Additional Experiments**

We thank the reviewers for the time and effort dedicated to evaluating our paper. Some reviewers requested: a) additional experiments on other tasks (beyond text classification) and b) stronger HPO/NAS baselines. In the following, we provide preliminary results that address both of these points. The final, statistically validated results will be included in the revised version. Other points raised by the reviewers will be addressed individually.

We evaluated LAMPS on machine translation using 38 language pairs from the OPUS Books corpus, optimizing for model size and validation loss. We report the average GPU-hours required to recover 99.9% of the optimal hypervolume, except by the `Exhaustive` collumn, which indicates the time to evaluate all candidates completely (just for reference). We additionally include [ASHA+EpsNet](https://arxiv.org/pdf/2106.12639), a stronger baseline used in HPO and NAS. LAMPS remains the best-performing method on most held-out datasets, with a few exceptions (mostly related to Russian translations, for some reason). Remarkably, LAMPS is comparable to Oracle in 78.9% of cases, a similar score to our original experiment with far fewer datasets (77.7%).

|Held-out Dataset | Oracle | LAMPS (ours) | ASHA+EpsNet | ZigZag | Random | Exhaustive|
|---|---|---|---|---|---|---|
| Helsinki-NLP/opus_books[de-en] | 41.24 ± 3.69 | 86.75 ± 1.16     | **42.20 ± 2.61** | 83.15   | _73.97 ± 6.39_ | 123.95 |
| Helsinki-NLP/opus_books[de-es] | 15.71 ± 0.44 | **16.27**        | _31.90 ± 5.04_   | 50.87   | 57.83 ± 4.19   | 73.64  |
| Helsinki-NLP/opus_books[de-fr] | 15.99 ± 0.21 | **15.87**        | _29.91 ± 1.97_   | 63.99   | 57.15 ± 7.35   | 92.96  |
| Helsinki-NLP/opus_books[de-it] | 18.11 ± 0.95 | **18.39**        | _33.73 ± 1.71_   | 55.17   | 63.42 ± 4.40   | 77.12  |
| Helsinki-NLP/opus_books[de-nl] | 10.96 ± 0.23 | **10.89**        | _17.82 ± 1.83_   | 31.61   | 36.09 ± 2.23   | 45.40  |
| Helsinki-NLP/opus_books[de-pt] | 1.42 ± 0.08  | **1.33 ± 0.03**  | _3.06 ± 0.19_    | 4.50    | 5.25 ± 0.28    | 6.24   |
| Helsinki-NLP/opus_books[de-ru] | 3.67 ± 1.14  | _16.62 ± 1.01_   | **12.97 ± 3.25** | 25.09   | 26.96 ± 5.71   | 51.17  |
| Helsinki-NLP/opus_books[en-es] | 55.76 ± 3.72 | **55.67 ± 4.30** | _108.29 ± 9.02_  | 172.28  | 184.54 ± 14.77 | 245.12 |
| Helsinki-NLP/opus_books[en-fi] | 2.00 ± 0.43  | **3.38**         | _5.36 ± 0.18_    | 6.10    | 9.09 ± 1.11    | 12.58  |
| Helsinki-NLP/opus_books[en-fr] | 54.80 ± 1.24 | **56.98**        | _96.95 ± 23.38_  | 223.08  | 167.94 ± 26.35 | 314.04 |
| Helsinki-NLP/opus_books[en-it] | 19.98 ± 0.36 | **19.75 ± 0.59** | _37.66 ± 2.99_   | 62.09   | 69.65 ± 4.94   | 89.49  |
| Helsinki-NLP/opus_books[en-nl] | 31.19 ± 1.02 | **31.05 ± 2.33** | _50.10 ± 3.55_   | 78.36   | 94.96 ± 4.98   | 111.73 |
| Helsinki-NLP/opus_books[en-no] | 2.68 ± 0.30  | **2.85**         | _4.62 ± 0.92_    | 8.30    | 7.28 ± 0.71    | 11.81  |
| Helsinki-NLP/opus_books[en-pl] | 2.14 ± 0.10  | **2.14**         | _3.67 ± 0.13_    | 6.15    | 6.09 ± 0.57    | 8.96   |
| Helsinki-NLP/opus_books[en-pt] | 2.18 ± 0.09  | **1.30 ± 0.02**  | _3.06 ± 0.26_    | 4.44    | 4.82 ± 0.22    | 5.80   |
| Helsinki-NLP/opus_books[en-ru] | 2.94 ± 1.06  | _12.66_          | **6.72 ± 2.43**  | 23.67   | 26.64 ± 5.98   | 47.85  |
| Helsinki-NLP/opus_books[en-sv] | 3.73 ± 0.15  | **3.22**         | _4.73 ± 0.69_    | 8.26    | 7.37 ± 0.59    | 11.52  |
| Helsinki-NLP/opus_books[es-fi] | 3.40 ± 0.09  | **3.59**         | _5.40 ± 0.49_    | 8.80    | 9.00 ± 0.73    | 12.18  |
| Helsinki-NLP/opus_books[es-fr] | 28.06 ± 1.31 | **28.56 ± 0.34** | _48.89 ± 4.08_   | 99.39   | 82.61 ± 11.75  | 143.97 |
| Helsinki-NLP/opus_books[es-it] | 21.14 ± 0.77 | **21.69**        | _34.45 ± 2.87_   | 59.67   | 68.76 ± 3.95   | 84.95  |
| Helsinki-NLP/opus_books[es-nl] | 26.07 ± 0.13 | **25.95**        | _45.90 ± 5.76_   | 70.69   | 80.08 ± 5.17   | 97.79  |
| Helsinki-NLP/opus_books[es-no] | 3.12 ± 0.13  | **3.41**         | _4.20 ± 0.69_    | 8.87    | 8.01 ± 1.10    | 12.41  |
| Helsinki-NLP/opus_books[es-pt] | 1.67 ± 0.15  | **1.29**         | _3.11 ± 0.24_    | 4.33    | 4.79 ± 0.32    | 6.31   |
| Helsinki-NLP/opus_books[es-ru] | 4.02 ± 1.15  | _15.31_          | **6.52 ± 2.20**  | 25.77   | 22.23 ± 5.80   | 50.55  |
| Helsinki-NLP/opus_books[fi-fr] | 2.95 ± 0.25  | **5.96**         | _8.93 ± 0.29_    | 10.99   | 7.38 ± 0.83    | 11.22  |
| Helsinki-NLP/opus_books[fi-no] | 4.47 ± 0.51  | **3.06 ± 0.07**  | _5.35 ± 0.38_    | 8.79    | 9.50 ± 0.58    | 12.13  |
| Helsinki-NLP/opus_books[fi-pl] | 2.39 ± 0.09  | **2.40 ± 0.01**  | _4.33 ± 0.02_    | 7.47    | 7.55 ± 0.80    | 10.59 |
| ... | ...  | ...  | ... | ... | ... | ... |

(Table truncated due to characters limitations in the response. Refer to the revised version for the complete results.)

---

> ### Author Response · Authors · 2025-12-02
> **Comparison against classic (single-objective) HPO methods**
>
> Dear all,
>
> One of the reviewers requested comparisons against classic single-objective HPO methods, such as Successive Halving (SHA), Hyperband (HB), and BOHB. Although our primary focus is the multi-objective setting, LAMPS applies naturally to the single-objective case without any modification, so we evaluated it on the same machine-translation benchmark used before, now optimizing only the validation loss and measuring the time required to reach 99% of the optimal loss. This enables a clean, direct and fair comparison with standard HPO baselines.
>
> In this setting, LAMPS delivers even stronger results when compared with the 2D and 3D cases, achieving a win rate of 35/38 (92.1%). SHA emerges as a surprisingly competitive baseline, yet still fails to reach the 99% threshold on many held-out datasets. Note that the strongest baseline in our main (multi-objective) experiments, ASHA+EpsNet (MO-ASHA), is itself a multi-objective extension of SHA, underscoring the relevance of this comparison.
>
> Together, these additional results, combined with the evidence presented earlier, provide a clearer picture of the capabilities of LAMPS. They show that, within the problem setting we study (pure model selection under expensive candidate evaluations) LAMPS consistently offers the most efficient and reliable search performance among the methods we compared.
>
> We thank the reviewers once again for their thoughtful feedback, which helped us substantially strengthen the case for LAMPS.
>
> | Held-out Dataset | Oracle | LAMPS (ours) | SHA | Hyperband | BOHB | ZigZag | Random | Exhaustive |
> |---|---|---|---|---|---|---|---|---|
> | de-en | 5.5  | _5.5_         | **4.4** | 29.2 ± 10.6 | 20.0 | 11.7 | 37.4 ± 11.0 | 123.9 |
> | de-es | 3.3  | **3.3**       | _4.1_   | 24.2 ± 11.2 | 16.3 | 20.5 | 31.3 ± 9.5 | 73.6 |
> | de-fr | 4.3  | **4.3**       | _5.2_   | 19.5 ± 1.8 | 20.0 | 9.5 | 35.1 ± 8.7 | 93.0 |
> | de-it | 4.1  | **4.1**       | _4.9_   | 20.0 ± 5.1 | 16.9 | 21.6 | 35.7 ± 9.7 | 77.1 |
> | de-nl | 2.6  | **2.6**       | _3.0_   | 11.6 ± 2.8 | 9.9 | 12.6 | 21.7 ± 5.5 | 45.4 |
> | de-pt | 0.5  | **0.5**       | _0.6_   | 2.5 ± 1.0 | 1.9 | 2.0 | 3.5 ± 0.6 | 6.2 |
> | de-ru | 0.5  | **4.7**       | -       | 18.7 ± 10.8 | 9.4 | 25.2 | 21.2 ± 6.1 | 51.2 |
> | en-es | 10.7 | **10.7**      | _13.2_  | 59.7 ± 9.8 | 52.7 | 39.2 | 97.7 ± 28.2 | 245.1 |
> | en-fi | 0.7  | **0.9**       | _1.0_   | 3.2 ± 0.2 | 2.9 | 1.1 | 5.3 ± 1.2 | 12.6 |
> | en-fr | 14.8 | **14.8**      | _18.0_  | 93.2 ± 51.4 | 76.1 | 89.4 | 158.8 ± 38.9 | 314.0 |
> | en-it | 4.4  | **4.4**       | _5.3_   | 50.3 ± 26.9 | 21.2 | 24.5 | 43.4 ± 12.8 | 89.5 |
> | en-nl | 5.2  | **5.2**       | _6.2_   | 27.7 ± 10.6 | 22.5 | 28.7 | 46.4 ± 11.9 | 111.7 |
> | en-no | 0.6  | **0.6**       | _0.8_   | 2.8 ± 0.3 | 2.8 | 2.1 | 5.7 ± 1.2 | 11.8 |
> | en-pl | 0.6  | **0.6**       | _0.7_   | 4.4 ± 1.8 | 2.4 | 2.9 | 4.2 ± 1.1 | 9.0 |
> | en-pt | 0.5  | **0.5**       | _0.6_   | 2.1 ± 0.3 | 2.0 | 2.1 | 3.2 ± 0.5 | 5.8 |
> | en-ru | 0.5  | **3.9 ± 0.9** | -       | 10.4 ± 2.2 | _8.7_ | 23.8 | 23.7 ± 6.8 | 47.8 |
> | en-sv | 0.6  | **0.7**       | _0.9_   | 3.8 ± 1.6 | 2.9 | 1.1 | 4.4 ± 1.2 | 11.5 |
> | es-fi | 0.8  | **0.8**       | _1.0_   | 3.1 ± 0.6 | 3.0 | 1.2 | 4.9 ± 1.1 | 12.2 |
> | es-fr | 6.1  | **6.1**       | _7.6_   | 30.4 ± 1.5 | 31.6 | 15.6 | 45.8 ± 16.2 | 144.0 |
> | es-it | 4.1  | **4.1**       | _5.0_   | 20.0 ± 4.4 | 19.7 | 16.4 | 35.0 ± 8.3 | 85.0 |
> | es-nl | 5.2  | **5.2**       | _6.1_   | 38.4 ± 19.5 | 21.8 | 29.0 | 47.9 ± 11.8 | 97.8 |
> | es-no | 0.7  | **0.7**       | _0.9_   | 4.8 ± 1.8 | 3.1 | 3.7 | 6.4 ± 1.6 | 12.4 |
> | es-pt | 0.4  | **0.4**       | _0.6_   | 2.1 ± 0.4 | 1.9 | 2.1 | 3.1 ± 0.6 | 6.3 |
> | es-ru | 0.5  | 5.6 ± 0.8     | **1.0** | 22.3 ± 14.4 | _8.0_ | 25.9 | 24.0 ± 6.1 | 50.5 |
> | fi-fr | 0.7  | **1.3**       | -       | 3.2 ± 0.6 | 2.9 | _1.4_ | 6.1 ± 1.6 | 11.2 |
> | fi-no | 0.6  | **0.8**       | **1.2** | 3.1 ± 0.6 | 2.9 | 1.3 | 4.5 ± 1.3 | 12.1 |
> | fi-pl | 0.6  | **0.6**       | _0.8_   | 3.0 ± 0.6 | 2.4 | 3.3 | 4.7 ± 1.4 | 10.6 |
> | fr-it | 2.2  | **2.2**       | _2.6_   | 10.3 ± 2.7 | 10.4 | 8.5 | 19.7 ± 5.3 | 43.7 |
> | fr-nl | 5.1  | **5.1**       | _6.1_   | 50.5 ± 27.5 | 25.0 | 32.5 | 51.6 ± 13.6 | 114.8 |
> | fr-no | 0.6  | **0.6**       | _0.8_   | 3.7 ± 0.7 | 2.8 | 2.2 | 5.3 ± 1.2 | 11.4 |
> | fr-pl | 0.5  | **0.5**       | _0.7_   | 3.9 ± 2.4 | 2.4 | 3.1 | 4.7 ± 1.3 | 9.5 |
> | fr-pt | 0.4  | **0.4**       | _0.5_   | 3.5 ± 1.3 | 1.9 | 2.0 | 3.1 ± 0.9 | 6.4 |
> | fr-ru | 0.3  | **2.1 ± 0.5** | -       | 8.3 ± 4.4 | _4.5_ | 12.5 | 11.1 ± 3.0 | 25.0 |
> | fr-sv | 0.6  | **0.6**       | _0.8_   | 3.9 ± 1.4 | 2.7 | 2.1 | 5.1 ± 1.2 | 11.4 |
> | it-nl | 1.0  | 8.1 ± 0.4     | -       | _4.6 ± 0.9_ | **2.8** | 6.2 | 6.6 ± 0.8 | 8.9 |
> | it-pt | 0.4  | **0.4**       | _0.5_   | 2.2 ± 0.6 | 1.9 | 2.0 | 3.3 ± 0.9 | 5.9 |
> | it-ru | 0.6  | **4.9**       | -       | 20.0 ± 15.1 | _8.8_ | 27.2 | 26.9 ± 6.0 | 54.4 |
> | it-sv | 0.7  | **0.7**       | _0.9_   | 2.7 ± 0.2 | 2.7 | 1.2 | 4.5 ± 1.3 | 11.5 |

---

### Author Response · Authors · 2025-11-25

Dear all,

We have updated the paper and implemented a substantial set of improvements based on your feedback. The main changes are:

1. Stronger multi-objective baseline: We now include MO-ASHA, a multi-objective extension of ASHA, used for HPO/NAS.
2. New machine translation experiments: We added evaluations on 38 language pairs using both two and three objectives, providing a more rigorous test of objective-agnosticism and scalability (Appendix).
3. Expanded text-classification benchmark: Three additional datasets (SMS Spam, IMDB, and Financial Phrasebank) were incorporated, leading to a more comprehensive evaluation.
4. The main experiments now optimize validation loss and model size (rather than training time). Training time is highly noisy and hardware-dependent, while model size provides a clearer, more reliable objective for real-world model selection
5. Enhanced Related Work: We added a dedicated paragraph discussing the relation between HPO/NAS methods and LAMPS, clarifying conceptual similarities and important differences.
6. Simplified training algorithm: We replaced Distral with vanilla multi-task RL (MTRL). Ablation studies showed that Distral yields only marginal gains, and the simpler MTRL adoption improves clarity and reproducibility.
7. Numerous clarifications and minor revisions throughout the text. The major edits are highlighted in blue in the updated manuscript.

---

With these changes, we believe we have addressed the core concerns raised by the reviewers, particularly:

- the absence of strong HPO/NAS baselines,
- the need for stronger evidence of objective agnosticism,
- and the importance of assessing scalability with additional objectives.

We thank the reviewers again for their constructive feedback and hope the revised version reflects these improvements clearly.

---

### Meta-Review · Area_Chair_9Aq7 · 2026-01-07

**Summary:**

All reviewers find the problem being addressed by the current study interesting and the proposed problem formulation and proposed approach novel and well-designed.
There have been concerns about the manuscript, where some of the main concerns include: insufficient comparison to existing multi-objective and single-objective baselines, doubts about several technical aspects of study requiring clarifications, and lack of comprehensive discussion regarding relevant prior work.
The authors have provided detailed point-by-point responses to the reviewers concerns, addressing them to some extent.

**Reviewer Concerns:**

In the revision and rebuttal, the authors have provided further discussions regarding relevant prior work as requested by the reviewers.
Additional multi-objective baseline was considered - i.e., MO-ASHA, among several approaches mentioned by one of the reviewers - where evaluation results show that the proposed method consistently outperforms this baseline.
Furthermore, comparison against several existing single objective baselines has been performed, demonstrating the benefits of the proposed method under single-objective cases as well.
Several important doubts initially raised by the reviewers have been also clarified by the authors, improving the overall presentation and clarity of the work.

**Reviewer Scores:**

All reviewers initially gave a rating of 4, where all of them were confident about their assessment.
Based on the authors rebuttal, additional evaluations, further comparisons against prior work, and the clarifications made throughout the manuscript, it is likely that the reviewers might have slightly increased their rating from the original score of 4 to an improved average score around 5.
This would place the manuscript around or just below the borderline, implying that the overall work has potential and meaningful merits but also room for further improvement and evaluation.

---

### Decision · Program_Chairs · 2026-01-26

Reject